# The Heterochromatin protein 1 is a regulator in RNA splicing precision deficient in ulcerative colitis

Jorge Mata-Garrido[1], Yao Xiang[1], Yunhua Chang-Marchand[1], Caroline Reisacher[1], Elisabeth Ageron[1], Ida Chiara Guerrera[2], Iñigo Casafont [3,4], Aurelia Bruneau[5,6], Claire Cherbuy[5,6], Xavier Treton[7,8], Anne Dumay[7], Eric Ogier-Denis [7,9], Eric Batsché[10], Mickael Costallat[10], Gwladys Revêchon [11], Maria Eriksson [11], Christian Muchardt [10] & Laurence Arbibe [1]✉

Defects in RNA splicing have been linked to human disorders, but remain poorly explored in inflammatory bowel disease (IBD). Here, we report that expression of the chromatin and alternative splicing regulator HP1γ is reduced in ulcerative colitis (UC). Accordingly, HP1γ gene inactivation in the mouse gut epithelium triggers IBD-like traits, including inflammation and dysbiosis. In parallel, we find that its loss of function broadly increases splicing noise, favoring the usage of cryptic splice sites at numerous genes with functions in gut biology. This results in the production of progerin, a toxic splice variant of prelamin A mRNA, responsible for the Hutchinson-Gilford Progeria Syndrome of premature aging. Splicing noise is also extensively detected in UC patients in association with inflammation, with progerin transcripts accumulating in the colon mucosa. We propose that monitoring HP1γ activity and RNA splicing precision can help in the management of IBD and, more generally, of accelerated aging.

Inflammatory bowel diseases (IBDs), including ulcerative colitis (UC) and Crohn's disease (CD), are chronic inflammatory gut disorders characterized by an uncontrolled inflammation leading to bowel damage. While susceptibility gene loci have been identified, genetic factors account for only a portion of overall disease variance, indicating a need to better explore interactions between genes and environment during the development of the diseases[1]. Epigenetics captures environmental stresses and translates them into specific gene expression patterns, prompting us to explore chromatin deregulations in the pathogenesis of IBD. In an earlier study, we have identified Heterochromatin Protein 1γ (HP1γ) as a regulator of inflammatory genes in response to enterobacteria[2]. Members of the HP1 family of proteins, that also includes HP1α and HP1β in mouse and human, are readers of the H3K9me2/3 histone modifications. They play key roles in formation and maintenance of heterochromatin, thereby participating in transcriptional gene silencing[3]. In parallel, HP1 proteins

[1]Université Paris Cité, INSERM, CNRS, Institut Necker Enfants Malades, F-75015 Paris, France. [2]Proteomics platform Necker, Université Paris Cité-Structure Fédérative de Recherche Necker, INSERM US24/CNRS UAR3633, 75015 Paris, France. [3]The Nanomedicine Group, Institute Valdecilla-IDIVAL, 39011 Santander, Spain. [4]Anatomy & Cell Biology Department, Faculty of Medicine, University of Cantabria, 39011 Santander, Spain. [5]Micalis Institute, Institut National de Recherche pour L'agriculture, L'alimentation et L'environnement (INRAE), AgroParisTech, Université Paris-Saclay, UMR1319, F-78350 Jouy-en-Josas, France. [6]Paris Center for Microbiome Medicine (PaCeMM) FHU, AP-HP, F-75571 Paris, France. [7]UMR-S 1149, Université Paris Cité, Inserm, Centre de recherche sur l'inflammation, équipe Inflammation intestinale, F-75018 Paris, France. [8]Paris IBD Center, Centre hospitalier privé Ambroise Pare-Hartmann, Neuilly, France. [9]INSERM 1242 and Centre Eugène Marquis, Rennes, France. [10]Sorbonne Université, Institut de Biologie Paris-Seine (IBPS), CNRS Unit of Biological Adaptation and Ageing (B2A), Epigenetics and RNA metabolism in human diseases, 75005 Paris, France. [11]Department of Biosciences and Nutrition, Center for Innovative Medicine, Karolinska Institutet, SE-141 57 Huddinge, Sweden. ✉e-mail: laurence.arbibe@inserm.fr

display RNA binding activity, as described in multiple species[4], and in mammals, in vitro studies have shown that HP1γ binds intronic repetitive motifs of pre-messenger RNA[5] to promote co-transcriptional pre-mRNA processing and alternative splicing[6–8]. Here, we show that, in the gut epithelium, HP1γ-mediated regulation of both transcription and RNA metabolism is required for gut homeostasis and important in the understanding of IBD.

## Results

### HP1γ inactivation triggers IBD-like traits

Our previously reported observations on the impact of HP1γ on the control of inflammation in the gut in response to bacterial infection[2] prompted us to examine expression of this protein in the context of chronic inflammation. To that end, we examined an available cohort of colonic biopsies in non-inflamed tissue from UC patients and from healthy individuals undergoing screening colonoscopies (Fig. 1a and Supplementary Data 1 *for a detailed description of the population*). Quantitative immunofluorescence (IF) showed a strongly decreased expression of HP1γ in the colonic epithelium of UC patients, as compared to healthy individuals (control patients) (Fig. 1a, b). Compromised HP1γ expression in association with UC was confirmed using the EXCY2 mouse model, which combines immune dysfunction (Il10 deficiency) and epithelium NADPH oxidase 1 (Nox1) deficiency[9]. At early stages, these mice exhibited a spontaneous chronic colitis that evolved into a colitis-associated dysplasia and adenocarcinomas. Reduced HP1γ expression in the epithelium prevailed at the initial chronic inflammatory stage (1–5 months aged), while at the cancer stage, the expression was recovered, in coherence with the reported increased detection of the Cbx-protein family members in various cancers[10,11] (Fig. 1c, d).

These observations, suggesting a role for HP1γ in chronic inflammation, prompted us to generate a Villin-creERT2:*Cbx3*[-/-] mouse model, allowing inducible inactivation of the *Cbx3* gene (encoding the HP1γ protein) in the epithelial lineage of the gut. In these mice, referred to as *Cbx3* KO mice, tamoxifen gavage resulted in complete depletion in the HP1γ protein in the epithelium of the tested tissues, although the depletion was accompanied by an up-regulation of the HP1α and HP1β isoforms (Supplementary Fig. 1), as previously reported[12]. Observation of these tissues by immunofluorescence staining did not indicate any change in marks of constitutive heterochromatin, including H3K9me3 and H4K20me3, and chromocenters visualized by DAPI staining appeared unaltered (Supplementary Figs. 1, 2).

Next-generation RNA-sequencing on purified epithelial cells from either crypts, villi, and colon in young adult mice (8–10 weeks aged) indicated extensive changes in the transcriptome landscape in the *Cbx3* KO mouse epithelium, with increased signature scores in pathways involved in lipid oxidation, symptomatic of oxidative damage (Supplementary Data 2). Moreover, we noted a marked decrease in expression of mRNAs coding for antimicrobial peptides (AMPs), including defensins, cathelicidins, and regenerating gene (Reg) type of AMPs (Supplementary Data 3 and Fig. 1e). In the colon, we further noted a substantial increase in inflammatory gene expression confirmed by Q-PCR (Fig. 1f, g and Supplementary Data 2).

Inflammation as well as alterations of AMP productions being both conducive of gut microbiome dysbiosis[13,14], we further characterized the fecal microbiomes in mice (n = 6 males and n = 8 females) via Illumina sequencing of the V3–V4 region of bacterial 16 S rRNA. In subsequent UniFrac principal coordinates analysis, *Cbx3* KO mice clustered away from the WTs, indicative of a shift in the microbial communities (Fig. 1h and Supplementary Data 4 for statistics details). This shift was exacerbated in females (p value = 0.006) as compared to males (p value = 0.015), although the bacterial communities were similar in the two sexes prior to *Cbx3* inactivation (Fig. 1h). The bacterial composition also remained unchanged in *Cbx3* fl/fl female mice treated with tamoxifen in the absence of the Cre recombinase, ruling

out an effect solely induced by tamoxifen administration (Supplementary Fig. 3).

As gender may influence IBD risk factors[15], we pursued separate analyses of male and female mice. In female mice, 110 Operational Taxonomic Unit (OTUs) were modulated in response to HP1 inactivation while in males, only 60 OTU covaried significantly with HP1 inactivation (Supplementary Data 5). In females particularly, we observed an overrepresentation of colitogenic bacteria such *as E. coli and Alistipes*, and a profound down-regulation of anti-inflammatory bacterial species such as the short-chain fatty acids (SCFAs) producer *Ruminococcaceae* (Supplementary Fig. 4), both phenomena being symptomatic of a dysbiotic microbiota reported in IBD[16].

Thus, UC is associated with a reduced expression of HP1γ in the gut epithelium, while inactivation of the cognate gene in the mouse gut results in features typical of gut homeostasis rupture. We concluded that in the colon epithelium, HP1 exerts protective functions, conveying to anti-inflammation and microbiota homeostasis.

### Impact on proliferative homeostasis and maturation

We next delineated the homeostatic functions played by HP1γ in the small intestine. The transcriptome analysis was indicative of a de-silencing of E2F target-genes upon inactivation of *Cbx3* (Supplementary Data 2), in agreement with its reported role in retinoblastoma (Rb)-mediated control of cell division[17]. Consistent with an effect of *Cbx3* inactivation on the cell-cycle, the proliferation marker Ki67 in the mutant mice was ectopically detected beyond the normal proliferative compartment, extending all along the crypts and at the base of the villus axes (Supplementary Fig. 5a). Furthermore, a 1 h pulse of the thymine analog 5-bromo-29-deoxyuridine (BrdU), marking cells in S phase, resulted in a frequent labeling of cells within the intestinal stem cell (ISC) compartment of *Cbx3* KO mice, as evidenced by the significant increase in detection of BrdU positive cells at 0 to +4 positions of the crypt (Fig. 2a, b). In control (Ctrl) mice, this labeling was predominantly confined to the immediate ISC progeny compartment i.e., the transit amplifying cells, while for the ISC, BrdU was poorly incorporated, consistent with the prolonged $G_1$ phase of stem cells[18]. Expansion of the stem-cell niche in the *Cbx3* KO mice was also documented by an enlarged area of detection for the Olfm4 marker of stemness (Supplementary Fig. 5b, c). Finally, ex vivo enteroid 3D matrigel cultures displayed accelerated bud formation when cells were collected from *Cbx3* KO mice (Supplementary Fig. 6a, b). However, upon prolonged culture, the bud per organoid ratio drastically dropped in *Cbx3* KO-derived organoids, possibly as a consequence of cellular exhaustion (Supplementary Fig. 6a, b). In the organoids, detection of cells in S-phase by EdU incorporation also revealed aberrantly positive cells along the villus axes, with EdU positive cells filling the lumen of the organoids (Fig. 2c and Supplementary Fig. 6c).

Along with this altered proliferative homeostasis, villi from mutant mice were subject to maturation defects, as illustrated by a GSEA analysis, showing a strong association between genes deregulated by *Cbx3* inactivation and genes associated with an intestinal stem cell signature, while RT-qPCR reactions revealed increased expression of the stemness markers Ascl2 and Olfm4 in the mutant mice (Fig. 2d, e). Along the villus epithelium of the *Cbx3* KO mice, we also noted expression of nucleolin, a marker of the nucleolus, and electron micrography studies provided evidence for the presence of canonical nucleolar structures at the upper part of the villi, including granular component, fibrillar center and dense fibrillar component (Fig. 2f, g). This was in sharp contrast with the expected decline of nucleolin expression along the crypt-villus axis (Fig. 2f, g), a consequence of the progressive dilution of ribosomes in the post-mitotic cell populations normally thriving on ribosomes inherited from the progenitor cells[19]. Ectopic production of nucleolar organelles was finally documented by an increased 18 s rRNA production observed at both crypts and villi, in association with an enrichment in genes involved in ribosomal biogenesis (Fig. 2h and Supplementary

 

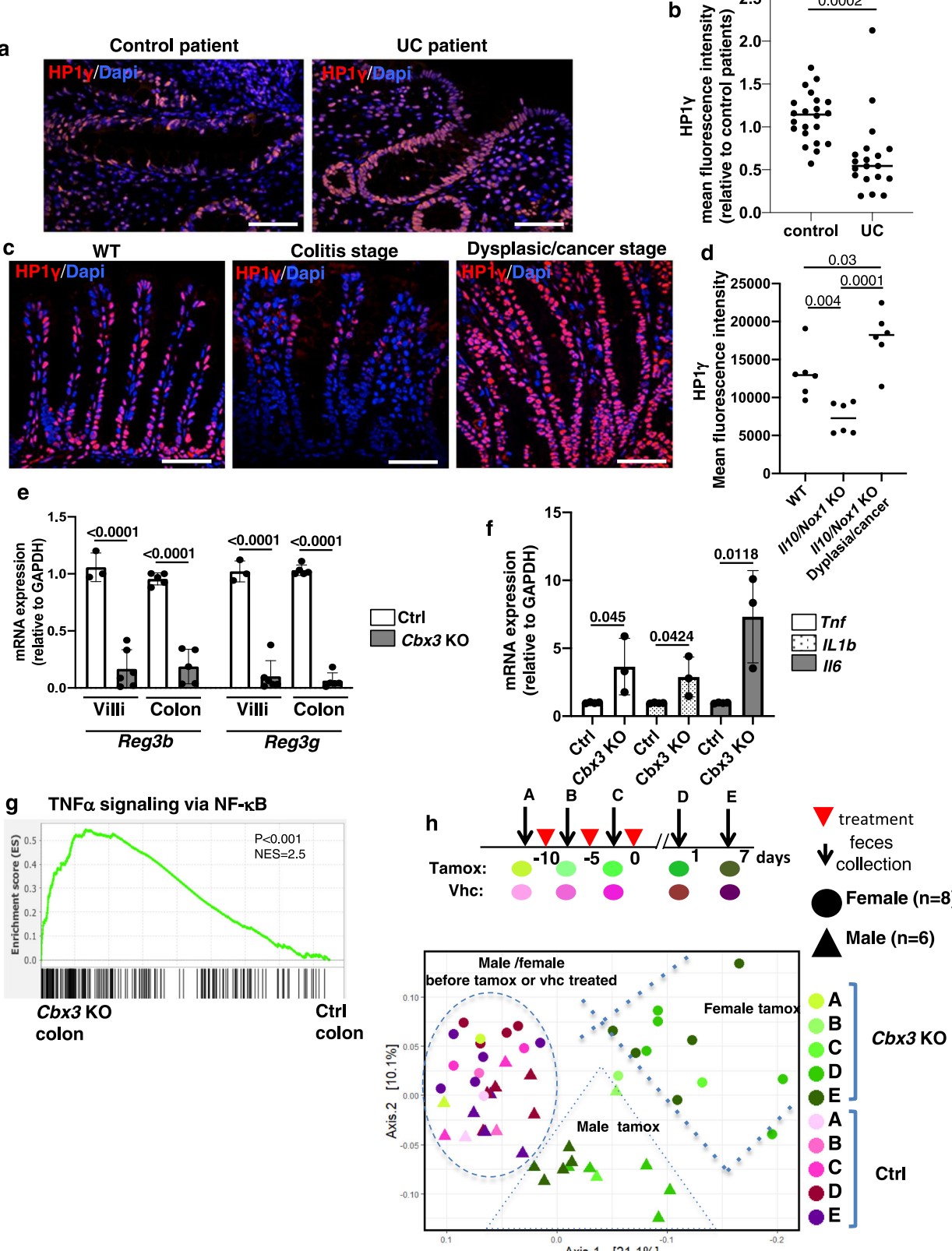

Fig. 5d). Thus, the homeostatic repression of nucleolar organelles occurring during epithelial cell maturation was lost. Of note, quantification of reads mapping to retrotransposons in the RNA-seq experiments documented that, unlike previously reported for inactivation of the H3K9 methyltransferase Setdb1 in the gut epithelium[20,21], inactivation of HP1γ did not result in increased expression of these normally heterochromatinized DNA repeats. Thus, de-silencing of rDNA and loss

of nucleolar repression along the villi in *Cbx3* KO mice did not occur in a context of global destabilization of heterochromatic structures (Supplementary Fig. 7).

Finally, the production of mature lineages in the *Cbx3* KO mice was affected on both absorptive and secretory lineages. Likewise, GSEA analysis of the RNA-seq data from both compartments showed alterations in the Paneth and enterocyte gene expression programs

**Fig. 1 | Cbx3 inactivation in the epithelium leads to gut homeostasis rupture.**
**a, b** HP1γ expression is affected in Ulcerative colitis (UC) patients: (**a**) Representative immunofluorescence in colonic tissue sections stained with anti-HP1γ antibody (red) and Dapi (blue) of colon sections (Scale bar: 50 μm) and in (**b**) ImageJ quantification of the mean fluorescence HP1γ signal intensity/section in control ($n = 10$) and UC patients ($n = 16$), expressed as relative value to control patients (**c, d**) Biphasic expression of HP1γ in the IL10/NOX1 KO mice model, **c** Immunostaining with anti-HP1γ antibody (red) and Dapi (blue) fluorescence and in (**d**) ImageJ quantification of the mean florescence Intensity. $n = 6$ mice for each group, Scale bar: 80 μm, (**e**) mRNA expression for *Reg3b* and *Reg3g* from villi and colon epithelia of Ctrl

(control) and *Cbx3* KO mice ($n = 3$–6 mice/group) (**f**) mRNA expression of *Tnf*, *Il1b* and *Il6* by RT-qPCR from colon epithelium of Ctrl (control) and *Cbx3* KO mice ($n = 3$–4 mice/group) (**g**) Significant enrichment of the colon *Cbx3* KO transcriptome with pro-inflammatory signature, Two-sided nominal *P* values were calculated by GSEA. (**h**) Temporal evolution of the Beta diversity in Villin-Cre *Cbx3* male and female mice. Sheme illustrating the time-course of fecal sample collection before (defining group A) and after treatments (Vhc or Tamox) (defining groups B–E), in female ($n = 8$) and male ($n = 6$) mice. Statistical analysis is provided in Supplementary Data 4. All Data are presented as the mean ± SEM; two-sided Student's *t* test (**b, d, e, f**). Source data are provided as a Source Data file.

(Supplementary Fig. 5e, f), and we noted a marked defect in the expression of lysozyme, a Paneth cell marker, and of Sucrase Isomaltase, an absorptive enterocyte differentiation marker (Fig. 2i, k). Overall, these data were indicative of a deregulation in the control of cell proliferation and in the production of mature lineages upon depletion of HP1γ.

## HP1γ limits usage of poor splice sites
We next sought to define functions for HP1γ on RNA homeostasis in the gut epithelium. Mass spectrometry revealed that HP1γ interactants were highly enriched in component of the spliceosome (Supplementary Fig. 8a, b and Supplementary Data 6). Among them, we identified members of the catalytic step 2 spliceosome (also called U2-type spliceosomal complex C) and splicing factors essential in splice-site recognition, such as Ser/Arg-rich (SR) proteins. Challenging some of these protein-protein interactions with RNAse did not suggest an implication of intervening RNA molecules, in agreement with earlier reports[22] (Supplementary Fig. 8c). This was consistent with studies on the role of HP1γ in the regulation of pre-mRNA splicing[6,22]. Likewise, analysis of the RNA-seq data with the Multivariate Analysis of Transcript Splicing (rMATS) pipeline confirmed that splicing was extensively impacted upon HP1γ inactivation. Significant variations in splicing included a trend toward increased intron retention in the small intestine but not in the colon, with increased and decreased inclusion of alternative exons were observed (Fig. 3a and Supplementary Datas 7–9, significant (FDR < 0.05) alternative splicing events in crypt, villus and colon epithelia). Some of these events were validated by RT-qPCR (Supplementary Fig. 9).

The absence of a defined pattern in the impact of HP1γ on splicing prompted us to examine whether these numerous splicing events would result from noisy splicing[23], as related in cancer and neurodegeneration[24–27]. To that end, we identified unannotated splice junctions present in only one of the two experimental conditions (either WT or *Cbx3* KO) for each tissue. Abundance of these junctions that we henceforth will refer to as "de novo" was significantly increased upon *Cbx3* inactivation (Fig. 3b). This increase was in a range similar to that observed in cerebellar Purkinje neurons depleted of *Rbm17*, an RNA binding protein reported to repress cryptic splicing usage[26] (Fig. 3b). RT-PCR experiments further documented increased usage of cryptic splice sites at exon or intron induced by HP1γ inactivation (Supplementary Fig. 10).

Evaluating the quality of the consensus splicing donors and acceptors with a dedicated software revealed that, in average, the de novo junctions scored lower that annotated junctions, but higher than random sequences (Fig. 3c). This suggested that inactivation of HP1γ promoted usage of poor consensus splice sites, and possibly increased the opportunity range of splicing. Examination of publicly available data identified several molecular partners of HP1γ that when inactivated, result in a significant increase in the splicing noise. These partners include PPL1[28], a peptidyl prolyl isomerase-like spliceosome component, ZC3H13, a zinc finger protein recruiting RNA methyl transferases involved in *N*6-adenosine methylation (m6A) modification[29,30], and the Ser/Arg-rich protein SRSF5[31] (Supplementary Fig. 11).

Finally, examining splicing noise induced by *Cbx3* inactivation on a gene-per-gene basis further documented that, at a large majority of genes not affected at their expression level, the number of active splice sites was significantly increased (Fig. 3d and Supplementary Data 10 for the list of genes). These genes included regulators of gut homeostasis. In particular, we noted that the *Pkm* gene, producing both Pkm1 and Pkm2 mRNAs by alternative splicing, the latter safeguarding against colitis[32], showed an extensive increase in de novo splicing in the colon (Fig. 3e, f, left panels). Likewise, the *Cdh1* gene encoding E-cadherin, essential for the epithelial barrier function[33], and subject to premature termination by alternative splicing[34], was affected by *Cbx3* inactivation at both crypts and colon (Fig. 3e, f middle panels). Finally, we observed a particularly strong impact at the *Lmna* gene with an average 9-fold increase in de novo junctions at the villi (Fig. 3e, f, right panel).

## Control of the progerin splice variant by HP1γ
The biological significance of splicing noise has been poorly characterized in mammals, prompting us to investigate whether the increased usage of de novo splice-junctions at the *Lmna* gene in villi lacking HP1γ could result in the production of progerin. The *LmnA* gene includes 12 exons and, by alternative splicing, it will produce both lamin A and lamin C mRNA. Occasionally, a rare splicing event will also result in the production of progerin, a truncated version of lamin A acting as a dominant negative protein isoform responsible for the Hutchinson Gilford Progeria Syndrome (HGPS)[35,36]. In this syndrome of premature aging, production of this splice variant is facilitated by a genomic mutation increasing usage of a progerin splice site that in normal cells is used at low yields upon usage of a poor-consensus splice site[37].

We thus applied RT-PCR and Taqman quantitative PCR to the detection of lamin A and progerin transcripts in the mouse gut epithelium (Supplementary Fig. 12a and Fig. 4a, b). End-point RT-PCR using primers nesting on exon 9 and 12 for *LmnA* showed evidence for progerin transcript detection in *Cbx3* KO but not in Ctrl mice (Supplementary Fig. 12a). Taqman quantitative PCR further documented a significantly increased occurrence of the progerin-specific splicing event upon *Cbx3* inactivation in the villus epithelium (Fig. 4b). Sequencing the PCR end-products confirmed that this splicing product was identical to that detected in G609G HGPS mice (Fig. 4c). The abundance of the splicing product remained however notably lower than observed in colon or small intestine epithelium from *Lmna* G609G HGPS mice, in coherence with the increased strength of the progerin 5'SS provided by the mutation[37] (Fig. 4b and Supplementary Fig. 12b).

We next used digital droplet PCR (ddPCR) to evaluate in each condition the Percent Spliced In (PSI) score of the progerin-specific splicing event, corresponding to the quantity of progerin isoform divided by the total quantity of canonical (lamin A) and progerin isoforms. In Ctrl crypts and villi, we detected an average of 415 ± 166 copies/ul and 0.16 ± 0.15 copies/ul of canonical and progerin-specific splicing products respectively, while, in the *Cbx3* KO, these figures raised to 1566 ± 725 copies/ul for lamin A and 2.8 ± 0.42 copies/ul for progerin (Supplementary Fig. 12c). As expected, progerin detection in cDNA derived from purified crypt epithelial cells of HGPS mice was

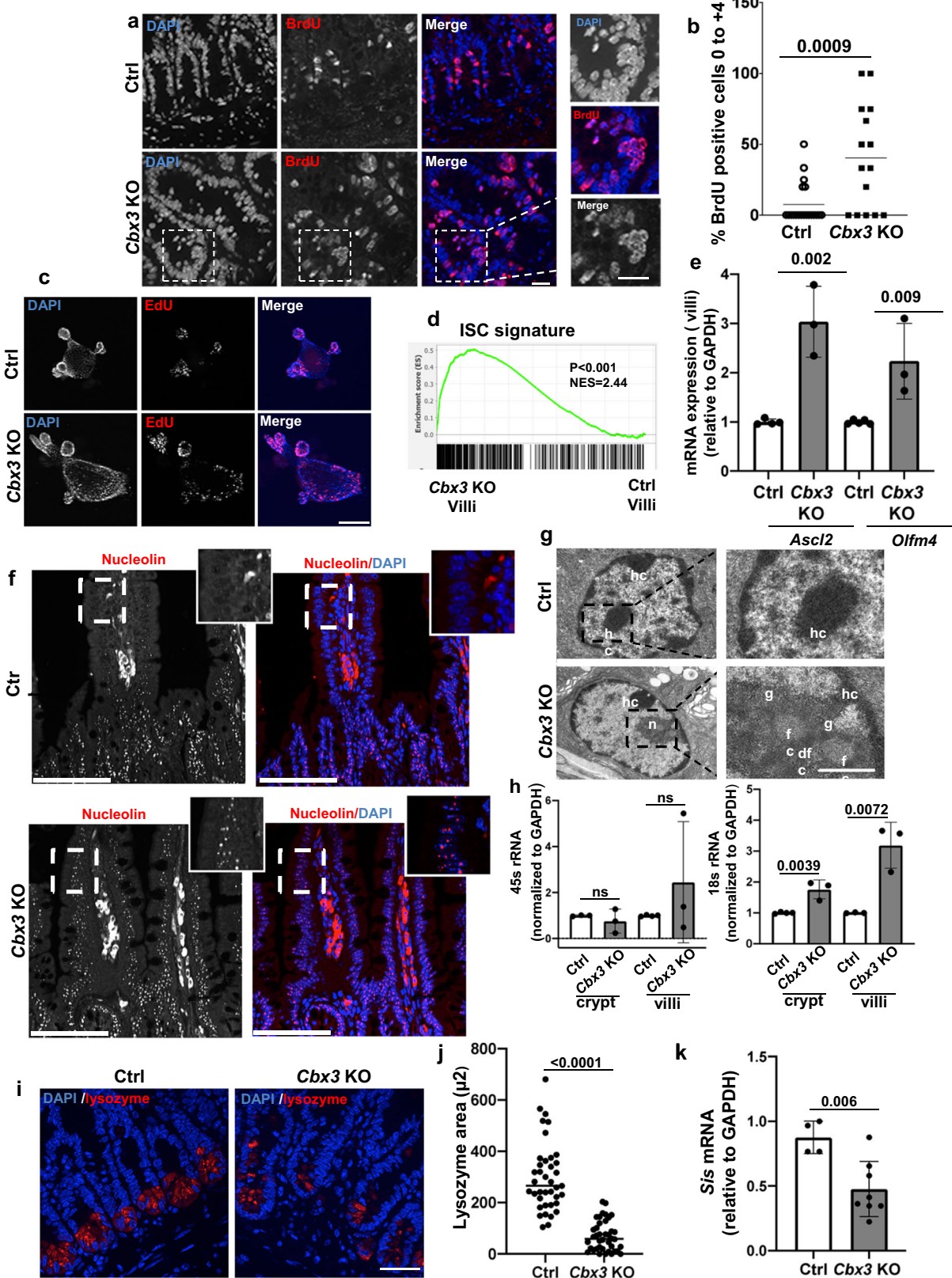

higher with 33.86 copies/ul and 11,8 copies/ul of lamin A. Translation into average PSIs showed an increase from 0.045 to 0.21 upon inactivation of *Cbx3*, documenting an increased occurrence of the progerin-specific splicing event that was not due to the increased expression of the *Lmna* gene in the mutant mice (Supplementary Fig. 12d and Supplementary Data 11). We next investigated whether the level of progerin transcript detected in the small intestine of the *Cbx3* KO mice

yielded production of progerin. Using a well-characterized anti-progerin monoclonal antibody (Supplementary Fig. 13), we readily detected progerin protein at both villus and crypt epithelia, as visualized by immunoblot (Fig. 4d). Of note, the absence of progerin signal in *Cbx3* fl/fl not harboring an inducible Cre-recombinase allowed to rule out an effect of the tamoxifen induction on progerin production (Fig. 4d). Accumulation of progerin protein in crypts and villi

**Fig. 2 | HP1γ controls epithelial proliferation and maturation in the small intestine.** (**a**) Immunostaining with anti-BrdU antibody (red) and Dapi (blue) of crypt ileal sections from Ctrl and *Cbx3* KO mice (Scale bar: 20 μm) and in (**b**) Quantification of the BrdU positive cells (red) vs Dapi (blue) at the stem cell compartment (0 to +4 position) n = 3 animals Ctrl, n = 20 sections; *Cbx3* KO n = 15 sections), (**c**) Representative image of co-staining EdU (red)/Dapi (blue) in organoids derived from Ctrl and *Cbx3* KO mice (Scale bar: 100 μm) in n = 3 mice/ group and quantification is provided in Supplementary Fig. 6c (**d**) Significant enrichment of the villi *Cbx3* KO transcriptome with the Lgr5+ intestinal stem cell signature (ISC) (Munoz et al., 2012), Two-sided nominal *P* values were calculated by GSEA, (**e**) mRNA expression of *Olfm4* and *Ascl2* by RT-qPCR from villi epithelium of Ctrl and *Cbx3* KO mice (n = 3–4 mice/group) (**f**) Representative immuno-fluorescence with nucleolin antibody, n = 6 mice/group (Scale bar: 100 μm, 25 uM in the insert) (**g**) Representative Transmission electron microscopy (n = 3 mice per condition) characterizing the nucleolar structure at the upper part of the villi: in (**g**-Left), Heterochromatin (hc) was observed in the nucleus of ctrl mice and in *Cbx3* KO mice canonical nucleoli (**n**) were detected, scale bar = 5 μm, in (**g**-right): Magnification showing the area of interest with a canonical nucleolus in the *Cbx3* KO mouse (g: granular component; fc: fibrillar center; dfc; dense fibrillar component) scale bar = 2 μm, (**h**) rRNA 45 S and 18 S expression levels in both crypt and villi in the *Cbx3* KO mice (n = 3–4mice/group), (**i**) Representative immunofluorescence with anti-lysozyme antibody (red) and Dapi (blue) at the ileal crypt (Scale bar: 20μm) and (**j**) Quantification of the lysozyme expression area. (n = 6 mice/group, with a total of n = 40 field/conditions) (**k**) mRNA expression of *Sis* (sucrase isomaltase) in the villi epithelium (n = 4–8 mice). All Data are presented as the mean ± SEM; two-sided Student's *t* test (**b, d, e, h, j, k**). Source data are provided as a Source Data file.

was further documented by immunocytochemistry (Supplementary Fig. 14). This approach revealed that progerin was principally detected at the immediate progeny of ISC (above + 4 position, Fig. 4e, f), thus evidencing for a gradient of progerin expression along the crypt-villus axis in *Cbx3* KO mice. In vitro crispr/Cas9-mediated *Cbx3* knockout (KO) in enterocytic TC7 cells confirmed the production of progerin transcripts, and accumulation of nucleocytoplasmic progerin protein (Supplementary Fig. 15a, b).

Accumulation of progerin chiefly disrupts the structure of the nuclear lamina[38]. We thus examined whether toxicity to the nuclear lamin A was detected upon *Cbx3* inactivation. Observation of the small intestine of *Cbx3* KO mice revealed a misshaping of the nuclear envelope in a fraction of the epithelial cells upon laminB1 immunostaining (Fig. 4g, h). Similar defects were also observed in vitro upon crispr/Cas9-mediated *Cbx3* knockout (KO) in the TC7 enterocytic cell line, resulting in deep invaginations of the nuclear membrane detectable by electron microcopy, reflecting a laminopathy phenotype (Supplementary Fig. 15c, d). Overall, these data showed that the HP1γ defect increased the opportunity range of lamin A mRNA splice variants, thus leading to the production of progerin in the absence of HGPS mutation in the gut epithelium.

**A non-canonical splicing signature in UC**

The reduced levels of HP1γ associated with UC (Fig. 1a, b) led us to search for an increased splicing noise in UC patients. To investigate this, we mined RNA-seq data from 206 young UC patients and 20 matched healthy donors from the large multicenter inception treatment-naïve UC cohort PROTECT[39]. This data set was selected because it was well controlled and had a depth of sequencing sufficient for splicing analysis. Quantifying de novo junctions revealed a highly significant increase (*P* Value = $10^{-59}$) in splicing noise in the UC patients, as compared to the healthy donors (Fig. 5a). Comparing UC patients displaying a de novo junction levels in the upper quartile to those with levels in the lower quartile, (see schematic in Fig. 5b), further revealed that high splicing noise correlated with increased activity of genes associated with the inflammatory response and with reduced expression of mitochondrial genes, both being linked to disease severity in the initial study[39] (Fig. 5b, c and Supplementary Data 12 including UC patient description and functional annotation enrichment analyses). Consistent with this, the histologic severity score was higher (computed based, *p* Value = 0.03). Splicing noise also appeared to have predictive value for mucosal healing after 4 weeks (4H) of medication (defined as fecal calprotectin <250 mcg/gm), as high levels of 4H fecal calprotectin were more frequent in the upper quartile (80% vs 57% in the lower quartile), and patients in remission were fewer (43% vs 67% in the lower quartile – Supplementary Fig. 16a, b). We further examine splicing noise on a *per* gene basis, as in the *Cbx3* KO mice. Out of 2743 genes under scrutiny, selected as being expressed in the biopsies (minimum 10 RNA-seq reads per gene) and unaffected by UC (less than twofold change in expression between healthy donors and UC patients), 415 genes (15%) displayed an increase in de novo junctions

>1.5-fold (*p* Val < 0.01) in UC patients, while only 11 genes (0.4%) showed reduced levels of these junctions (Fig. 5d and Supplementary Data 13). These genes included *LMNA* (Fig. 5e), prompting us to examine the production of progerin in a cohort of colonic biopsies. Total RNAs were extracted from colon biopsies of UC patients (n = 19) and compared to healthy individuals (n = 17), or Crohn's disease (CD) patients (n = 16) with colonic involvement (Supplementary Data 14 for detailed description of the population). Lamin A and progerin mRNA expression levels were then determined by taqman assays, including verification of PCR end-products by sequencing. In UC patients, levels of progerin-specific splicing events were significantly up-regulated (*p* Val = 0.0002) (Fig. 5f, g), while production of the lamin A splicing product was not (Fig. 5h). No correlation was detected between progerin mRNA expression and patient age (Supplementary Fig. 16c). Inversely, in CD patients, progerin-specific splicing was unaffected (Fig. 5f), while the lamin A-specific splicing product was up-regulated (*p* Val = 0.016) (Fig. 5h), this increase mainly concerning the younger CD patients <45 years old, as shown by linear regression analysis (Supplementary Fig. 16d, e).

Thus, these observations identify an extensive deregulation of splicing precision in UC. Progerin transcript emerges as a potential marker of UC and of this increased splicing noise. Finally, our observations stress the predictive value of the *Cbx3* KO mouse in modeling this disease.

## Discussion

While the role of RNA metabolism has been extensively described in neurodegenerative, cardiac, and premature aging diseases[40], it was largely unexplored in intestinal disorders. The present work defines HP1γ as a protein safeguarding RNA splicing accuracy in the gut epithelium, limiting the impact of naturally occurring non-canonical splicing events. Non-canonical splicing activity is frequently referred to as "splicing noise"[23]. While in non-mammalian animal models, noisy splicing may promote diversity in mRNA splice variants, with possible physiological functions[41,42], its significance in mammals remains elusive. In the latter, pervasive erroneous splicing activity may be a driver of genome evolution[43], but it has also been linked to human pathology, including cancer and neurogenerative diseases involving global activation of noncanonical splicing sites[25–27,44]. Only few RNA binding proteins, including Rbm17, hnRNP and TP43[26,27], were shown to dampen cryptic splicing globally. Based on the present study, HP1 can now be added to this list, with a fundamental role in the gut epithelium. The effect of HP1γ deficiency on splicing was sufficient to induce the production of progerin, a normally repressed splice variant of the *Lmna* gene in the absence of HGPS mutation. More generally, the effects of HP1γ inactivation on splicing was accounted for by an apparently general reduction in the precision of splicing. This was documented by the RNA-seq data, in which junction reads were increasingly detected at non-annotated splice junctions, and at sites with very poor donor or acceptor consensus sequences. The effect was also observable by semi-quantitative PCR, yielding a complex set of products in the

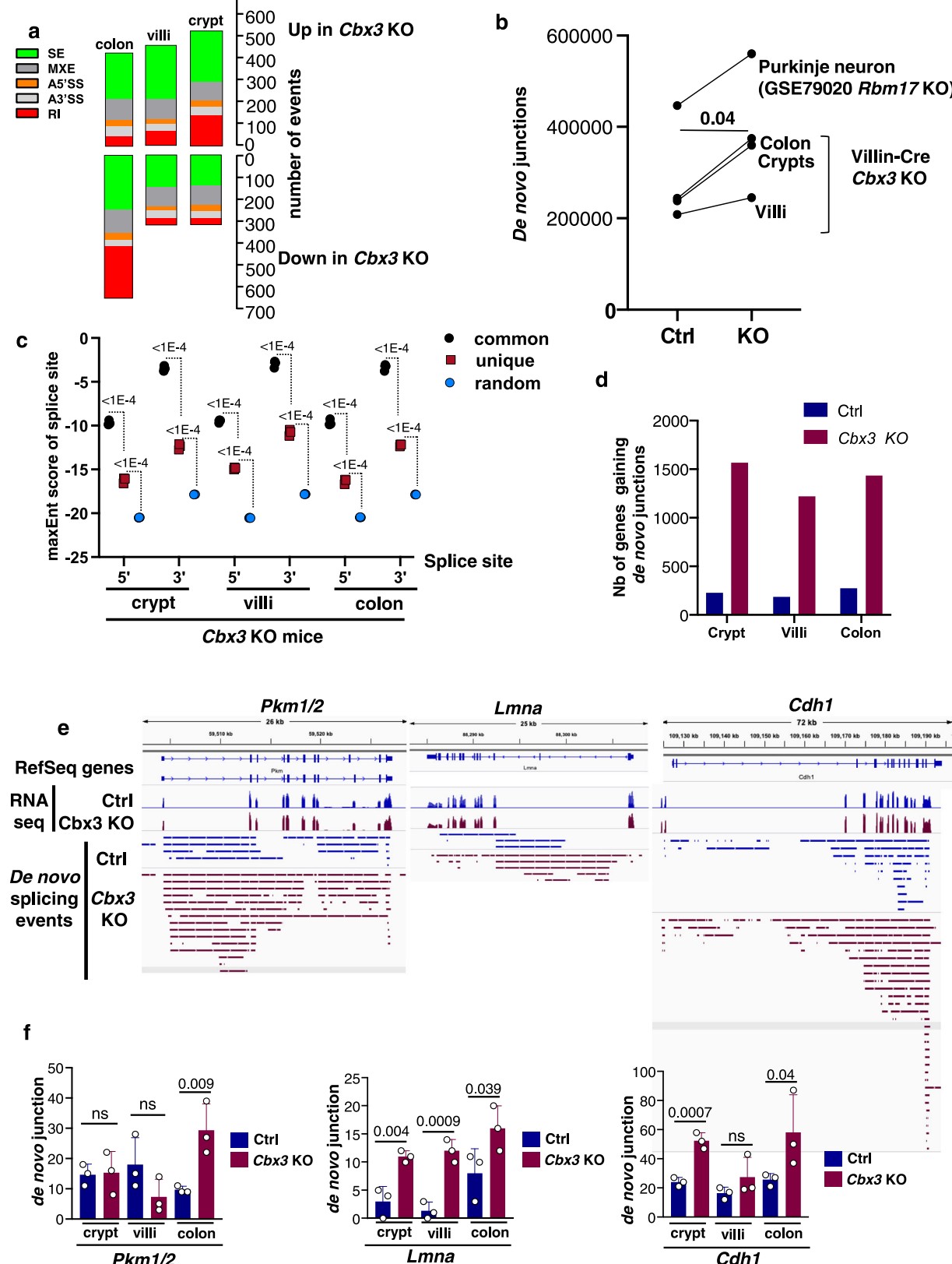

mutant mice with primers producing a single species from the wild-type tissues. While this decreased stringency in splice site selection may have multiple causes, including a possible uncoupling between transcription and splicing, we note that our proteomic approach to HP1γ molecular partners detected numerous splicing factors ensuring the usage of only genuine splice sites. These included components of

the major (U2-dependant) spliceosome as well as auxiliary splicing factors essential for spliceosome recognition of consensus sequences. Examining available knock-out data for some of these splicing factors, we found that their inactivation resulted in a splicing noise similar to that observed upon inactivation of *Cbx3*. The examined splicing factors included the Ser/Arg-rich SRSF5 previously shown to reduce

**Fig. 3 | Impact of HP1γ deficiency on RNA splicing in vivo. (a)** Types and quantity of splicing alterations detected in the colon, villus, and crypt epithelia by rMATS. Types of splicing alterations are color coded as indicated. SE: skipped exons, MXE: mutually exclusive exons, A3'SS and A5'SS: alternative 3'/5' splice sites, RI: retained introns. **(b)** Quantification of de novo junctions in control (Ctrl) and *Cbx3* KO (KO) conditions in crypts, villi and colon epithelia after consolidating the 3 RNA-seq replicates from either ctrl or *Cbx3* KO mice in each tissue. De novo junctions were defined as junctions not annotated in the mm9 version of the mouse genome and not present in both WT and mutant samples. Indicated *p* value was calculated with a paired two-sided Student's *t* test, *n* = 3 (crypts, villi and colon). Quantification of de novo junctions in mouse Purkinje neurons either WT or inactivated for *Rbm17* is shown as a positive control. **(c)** Consolidated maxEnt score of the de novo sites identified with the approach described in **(b)**, compared to the score obtained with annotated junctions, and to those obtained with randomly selected sequences (black and blue, respectively). Values shown are from *Cbx3* KO in crypts, villi, and colon (*n* = 3 mice, *p* value were calculated with ordinary one-way Anova). **(d)** De novo junctions were quantified at genes that, transcriptionally, were affected less than twofold by *Cbx3* KO inactivation. For each indicated condition, we counted genes gaining de novo junction twofold or more (*p* value < 0.05). **(e)** De novo junctions in colon (*Pkm* and *Cdh1*) or villi (*Lmna*) were visualized with a genome browser in the neighborhood of the indicated genes. Each bar represents a "de novo" donor/acceptor couple, spanning from the 5' to the 3' splice site. **(f)** Bar graphs show the number of de novo junctions detected at the indicated genes at crypts, villi, and colon (*n* = 3 mice for each condition, data are presented as the mean ± SEM; Indicated *p* values were calculated with a two-sided Student's *t* test). Source data are provided as a Source Data file.

progerin production by redirecting splicing toward lamin A 5'SS utilization[37,45], PPL1, a peptidyl prolyl isomerase-like spliceosome component whose inactivation leads to splicing alterations with aberrant use of weak splicing sites[28], and also ZC3H13, a zinc-finger protein regulating levels of RNA N6-methyladenosine[29], a modification involved in various aspects of RNA metabolism, including RNA splicing[46–48]. We thus propose a model in which HP1γ reduces splicing noise by favoring recruitment to the chromatin-template of proteins involved in the precision of splicing decisions. The reliance of HP1γ on auxiliary regulators of splicing may also be illustrated by its regional impact on splicing, possibly as a function of the splicing regulators available in each tissue or cell type. In particular, we noted a gradient of progerin expression along the crypt-villus axis, leaving out the ISC compartment. The latter suggests a modulation of splicing possibly linked to the cellular developmental state, a notion further supported by the HGPS-iPSCs model, in which progerin was only observed upon cell commitment[49]. While our data mainly points toward an impact of HP1γ via spliceosome recruitment, an involvement for HP1γ in the degradation of defective splicing products may also be considered, as suggested by the implication of the yeast HP1 homolog in escorting pericentromeric transcripts to the RNA decay machinery[50].

Before the role of HP1 was associated with splicing, this protein was mainly known as a mediator of heterochromatin-dependent silencing, repressing transcription at repeated DNA sequences, including rDNA loci, and at a subset of gene promoters[2,17,51–53]. Our inactivation of HP1γ in the mouse gut clearly demonstrated that HP1-dependent transcriptional repression is essential for the biology of the intestine, participating in the silencing of genes controlling proliferation and inflammation. These experiments also reveal a role for HP1γ at rDNA loci, repressing ribosomal genes during maturation of epithelial cells. However, unlike what was previously observed upon inactivation of the H3K9 methyltransferase Setdb1 in the gut[20,21], or the triple knock-out of HP1α, HP1β and HP1γ in liver[54] (Supplementary Fig. 7), inactivation of HP1γ did not affect silencing of endogenous retroviruses (ERV), normally keep in check by interspersed heterochromatic structures. Likewise, the overall organization of chromocenters was preserved upon HP1γ inactivation. However, the possibility of an impact of HP1γ deficiency on the integrity of the heterochromatin structures without global loss of heterochromatin state can not be excluded at this point.

In conclusion, our observations illustrate the homeostatic functions of HP1γ involved in essential activities in both euchromatin and heterochromatin. The previously described role of HP1 in tissue longevity across the species[53,55] may be largely rooted in this polyvalence. While no human disease has been clearly linked to HP1 gene mutations, decreased HP1 expression was reported in syndromes of accelerated aging, including the Werner syndrome and HGPS[56–58]. We similarly identified a potent reduction of HP1 expression in UC patients as well as in mouse models relevant for this disease. From our *Cbx3* KO mouse model, we concluded that HP1γ deficiency may support aberrant pathways nurturing both inflammation and splicing defects at key homeostatic genes. Likewise, the increased detection of progerin in UC colon tissue prompts us to speculate that, alike what we observed in the *Cbx3* KO mice, the lamin A splice variant is only the "tip of the iceberg", indicative of a more extensive disturbance in RNA splicing precision endured by UC patients, and clearly illustrated by the PRO-TECT UC patient cohort. Still in its infancy with regard to intestine disease[59–61], the identification of mechanisms relying on RNA splicing dysfunctions in the chronically inflamed gut should transform our understanding of the functional decline in the gut epithelium of IBD patients.

## Methods

### Mouse models

C57BL/6 *Cbx3*[fl/fl] mice (provided by Dr Florence Cammas, IRCM, Montpellier, France) were crossed with Villin-CreERT2 mice (provided by Dr Cohen-Tannoudji, Institut Pasteur, Paris, France) to produce the C57BL/6 Villin-creERT2:*Cbx3*[-/-] mouse model (this study). C57BL/6 Heterozygote *Lmna*[G609G/G609G] mice (here referred as HGPS mice) were provided by Dr Maria Eriksson (Karolinska Institutet, Sweden). Male and female mice were fed by a standard diet (SD) rodent chow (2018 Teklad Global 18% Protein Rodent Diet, Harlan) composed of 60% carbohydrate fed *ad libitum*. The animals were kept under 12 h/12 h (8am/8 pm) light on/off cycle, a temperatura of 22 °C ± 2 and a humidity between 50–70%. Tamoxifen (0.5 mg/g) diluted in 20% clin-Oleic acid was administrated by oral gavage, at 3 doses every 5 days as described[62]. Control mice received 20% clinOleic acid alone by oral gavage. Additional controls using *Cbx3*[fl/fl] mice that do not express the Cre recombinase were identically treated with tamoxifen. All the experiments using Villin-creERT2:*Cbx3*[-/-] mouse model were performed with male of female mice 2–3 months of age. BrdU (Sigma) was injected intraperitoneally (i.p.) at 100 μg/g animal body weight, 1 h prior to sacrifice. Animal studies were approved by the ethical committee of Paris Descartes University (authorization number 17-022).

### Patients and biopsy specimens

All patients were followed in the Department of Gastroenterology (hôpital Beaujon, Paris). The protocol was approved by the local Ethics Committee (CPP-Ile de France IV No. 2009/17, and No2014-A01545-42) and written informed consent was obtained from all patients before enrollment. Colonic pinch biopsy allowing the extraction of the epithelial layer was obtained during endoscopic investigation in non-inflamed areas to allow a comparison with healthy tissues in the control population. Biopsies were performed in the right or the left colon or in case of cancer at least 10 cm away from the cancer site. For the immuno-histological study, 26 patients were included, with 10 healthy control and 16 UC patients (detailed of the population provided in Supplementary Data 1). For the transcriptional study, 56 patients were included, with 17 healthy controls, 19 UC and 16 CD patients with colonic involvement (detailed of the population provided in Supplementary Data 12). Total RNA was extracted from human biopsies with RNAble (Eurobio) and quantified using a ND-1000 NanoDrop

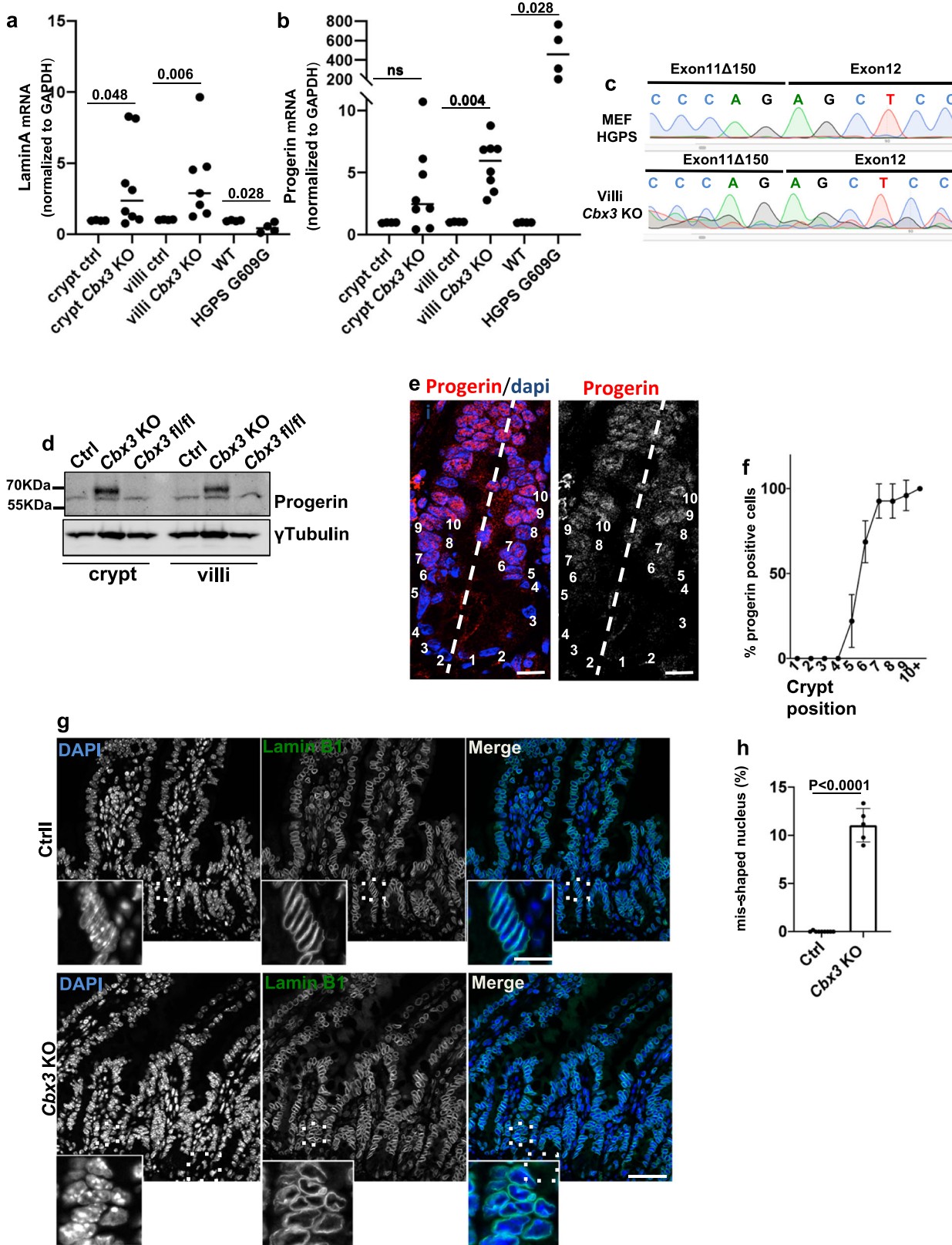

spectrophotometer (NanoDrop Technologies). Reverse transcription of total mRNA was performed using M-MLV RT (Invitrogen) according to the manufacturer's recommendations.

**Cell cultures and transfection**
Human dermal fibroblasts patients who carried the HGPS p.Gly608Gly mutation were obtained from the Coriell Cell Repository. Caco2 (TC7

cells, authenticated by ECACC) were used to generate the CRISPR/Cas9-mediated *Cbx3* cell line. The pSpCas9(BB)−2A-GFP (PX458) vector expressing Cas9 endonuclease (gift from Feng Zhang, Addgene plasmid # 48138) was linked with a single-guide RNA (sgRNA) designed specifically for *Cbx3* gene. Two Sequence guides (GAAGAAAATTTAGATTGTCC and GAATATTTCCTGAAGTGGAA) were defined by ZiFiT Targeter Version 4.2 software. Insertion of the sequence guide was

**Fig. 4 | Analysis of progerin production in the gut epithelium. (a, b)** Taqman Assays lamin A and progerin in ctrl ($n = 4$), *Cbx3* KO ($n = 8$), WT and HGPS G609G ($n = 4$ mice/group), Mann–Witney U test **(c)** Examples of sequencing data of the Taqman PCR end products in villi *Cbx3* KO samples, mouse embryonic fibroblast (MEF) derived from G609G mice used as control, **(d)** Immunoblot with anti-progerin monoclonal antibody in epithelium lysates derived from Ctrl, *Cbx3* KO mice and as internal control, *Cbx3 fl/fl* not expressing the Cre recombinase treated with tamoxifen. Experiment is representative of 2 independent repeats, with a total of $n = 4$ mice/group **(e)** Representative immunofluorescence with anti-progerin antibody (red) and Dapi (blue), scale bar: 20 µm and in **(f)** quantification provided by the percentage (%) of progerin expressing cells according to the position along the ileal crypt axis (*Cbx3* KO $n = 23$ sections, $n = 3$ mice). Data are presented as the mean ± SEM, **(g)** Immunofluorescence with anti-Lamin B1 antibody (green) marking the nuclear envelope (Scale bar: 50 µm, Insert: 15 µm) and in **(h)** quantification provided by the percentage of cells with misshaping nucleus, $n = 5$–9 mice in each group, counting in ctrl = 4757 nucleus and *Cbx3* KO = 29753 nucleus, respectively. Data are presented as the mean ± SEM, two-sided Student's *t* test. Source data are provided as a Source Data file.

performed in the BbsI restriction site of the PX458 vector and check by sequencing. Transfection in TC7 cells was performed by lipofectamine 2000 and singles clones for each sequence guide were selected by FACS according to the GFP signal. Transfections in HEK293T were performed by DNA–calcium phosphate precipitates using 5 µg of pLVX expression vector for Flag-V5 tagged Tomato, SRp20, RBM39 or TRA2β[63] and with 5 µg of pSG5-HA[64].

## Immunofluorescence

Intestine (ileum or colon) was collected and washed with PBS at 4 °C and cut in pieces about 5 mm. Intestinal fragments were fixed with formalin overnight at 4 °C. Once fixed, intestinal fragments were included in paraffin blocks. Paraffin sections were done in a microtome Leica RM2125 RTS, with a thickness of 4 µm. Subsequently, the deparaffinization and rehydration of the samples was carried out by immersion in Xylene (2 × 10 min), absolute ethanol 5 min, 90% ethanol 5 min, 70% ethanol 5 min and distilled water (2 × 5 min), all at R.T. Finally, the antigen was unmasked using the EDTA boiling technique for 30 min at 95 °C, followed by 20 min at R.T. All samples were sequentially treated with 0.1 M glycine in PBS for 15 min, 3% BSA in PBS for 30 min and 0.5% Triton X-100 in PBS for 2 h (mouse tissue). In case of nucleolin staining, tissue sections were incubated during 1 min at RT with proteinase K 0.05 mg/ml, followed by a wash with glycine 2 mg/ml during 15 min at RT. They were then incubated with primary antibodies overnight at 4 °C, washed with 0.05% Tween-20 in PBS, incubated for 1 h in the specific secondary antibody conjugated with Alexa 488 or Cy3 (Jackson, USA), 15 min with DAPI (1µg/ml), washed in PBS and mounted with the antifading medium VECTA-SHIELD® (Vector laboratories). Anti-HP1γ (2MOD-IG6, Thermo Scientific), HP1β (1MOD-1A9, Thermo Scientific), HP1α (2H4E9, Novus Biologicals), Ki67 (ab16667, Abcam), Olmf4 (D6Y5A, Cell Signalling), BrdU (MA3-071, Thermo Scientific), laminB1 (ab65986, Abcam), Progerin (13A4DA, sc-81611, Santa Cruz), γTubulin (4D11, Thermo Scientific), and Nucleolin (ab22758, Abcam) were used as primary antibodies. Nuclei were stained using 4, 6-diamidino-2-phenylindole (DAPI, 62248, Thermo Scientific).

Microscopy images were obtained with a ZEISS Apotome.2 (Zeiss, Germany), structured illumination microscope, using a 63× oil (1.4 NA) objective. To avoid overlapping signals, images were obtained by sequential excitation at 488 and 543 nm in order to detect A488 and Cy3, respectively. Images were processed using ZEISS ZEN lite software. The quantitative analysis of the immunofluorescence signal was performed on ImageJ. The values are represented with Mean fluorescence intensity, relatives to control samples.

## Tissue processing for intestinal epithelial cells isolation and organoids culture

The technique was adapted from *Nigro* et al., 2019[65]. Small intestine or colon were collected and washed with PBS at 4 °C and cut in pieces of about 5 mm in length. For epithelial isolation, intestinal fragments were incubated 30 min at 4 °C in 10 mM EDTA after which they were transferred to BSA 0,1% in PBS and vortexed 30−60 s. The supernatants (containing the epithelial cells) were filtered with a 70 µm cell strainer. At this step, crypts went through the cell strainer and villi were retained on it. Crypt and villus fractions were then centrifuged separately and

the pellets were frozen in liquid nitrogen until processed. The quality of the separation was accessed by the expression of selective expression of stemness markers in the crypts but not villi, as confirmed by RNA sequencing. For organoid production, crypt pellet was disaggregated and cultured in Matrigel as described[65]. EdU staining was performed using Click-iT™ EdU Cell Proliferation Kit for Imaging (Thermo fisher), following manufacturer indications.

## Transmission electron microscopy

In TC7 cells, Transmission Electron Microscopy (TEM) was performed by cryofixation/freeze substitution method. Cells were fixed overnight at 4 °C with 3.7% paraformaldehyde, 1% glutaraldehyde, in 0.1 M cacodylate buffer. After fixation, cells were pellet and contrasted with osmium tetroxide ($OsO_4$) and uranyl acetate. Then, cells were included in freeze-substitution medium during at least 3 days at −80 °C, dehydrated in increasing concentrations of methanol at −20 °C, embedded in Lowicryl K4 M at −20 °C and polymerized with ultraviolet irradiation. Ultrathin sections were mounted on nickel grids, stained with lead citrate and uranyl acetate and examined with a JEOL 1011 electron microscope.

Intestinal tissue fragments around 500 mm were fixed overnight at 4 °C with 3.7% paraformaldehyde, 1% glutaraldehyde in 0.1 M cacodylate buffer overnight at 4°. Small tissue fragments were washed in 0.1 M cacodylate buffer, dehydrated in increasing concentrations of methanol at −20 °C, embedded in Lowicryl K4 M at −20 °C and polymerized with ultraviolet irradiation. Ultrathin sections were mounted.

## Real time quantitative PCR and Droplet digital PCR (ddPCR)

Total RNA was extracted using Trizol (TR-118, Molecular Research Center, Inc.) following the manufacturer's instructions and DNAse treatment. RNA samples were quantified using a spectrophotometer (Nanodrop Technologies ND-1000). First-strand cDNA was synthesized by RT-PCR using a RevertAIT H Minus First Strand cDNA Synthesis kit (Thermo Scientific). qPCR was performed using the Mx3005P system (Stratagene) with automation attachment. For mouse progerin and lamin A semi-quantitative PCR, primers nested on exon 9 and 12 were used (F: GTGGAAGGCGCAGAACACCT R: GTGAGGGGGGAGCAGGTG), as reported[66]. For real-time PCR amplification, TaqMan assay were used with predesigned primers for mouse GAPDH (Mm99999915_g1), mouse lamin A that do not recognized progerin or DNA (Assay ID: APGZJEM), mouse progerin (F: ACTGCAGCGGCTCGGGG R: GTTCTGGGAGCTCTGGGCT and probe: CGCTGAGTACAACCT). These primer-probe pairs were further used for TaqMan ddPCR assay on cDNA samples. Droplet digital PCR (ddPCR) was performed on cDNA samples using the TaqMan probe based assay previously described for mouse lmna and progerin with ddPCR Supermix for probes (Bio-Rad). Droplets were generated from the reaction mixture using a droplet generator (Bio-Rad QX200). PCR amplification was performed as follow: after enzyme activation 95 °C for 10 min (1 cycle), 50 cycles of two steps during 20 s at 95 °C, 40 s at 60 °C, followed by enzyme deactivation 10 min at 98 °C. PCR droplets were measured into the automated droplet reader (Bio-Rad QX200). The raw transcript count per µl was calculated by Quantasoft software (Bio-Rad). For each sample and transcripts, at least two replicate experiments were performed.

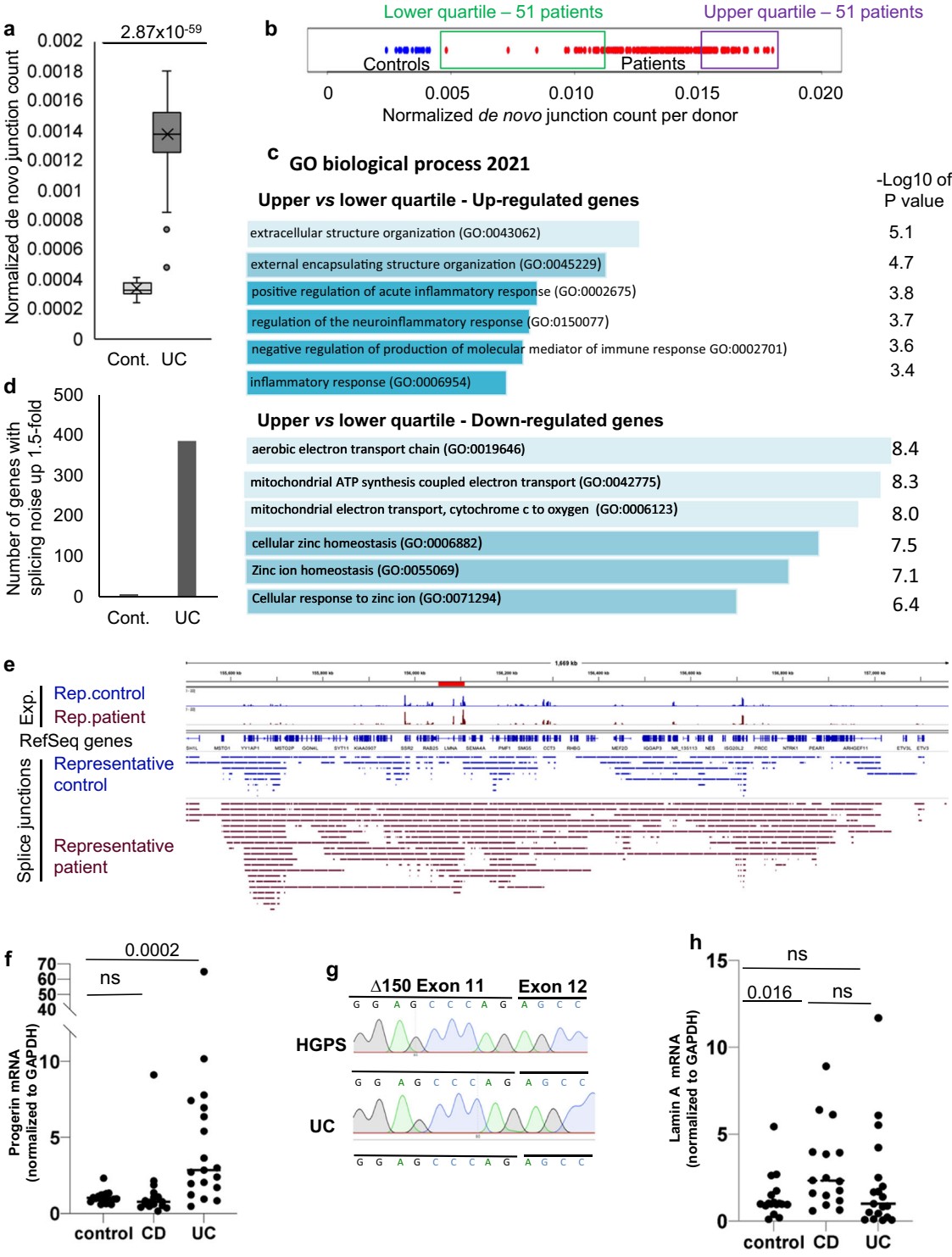

For detecting lamin A and progerin in human cDNAs, the primer-probe pairs were as followed: human lamin A (F: TCTTCTGCCT CCAGTGTCACG R: AGTTCTGGGGGCTCTGGGT and probe ACTCGCA GCTACCG), human progerin (F: ACTGCAGCAGCTCGGGG R: TCTG GGGGCTCTGGGCTCCT and probe CGCTGAGTACAACCT), human GAPDH (Hs00266705_g1). For SYBRGreen (Takara) based qPCR, the following primers have been used: mouse Ascl2 (F: GGT GAC TCC TGG TGG ACC TA; R: TCC GGA AGA TGG AAG ATG TC) mouse Olfm4(F:ATC AGC GCT CCT TCT GTG AT R: AGG GTT CTC TCT GGA TGC TG) mouse TNF-α (F: GATCTCAAAGACAACCAACATGTG R: CTCCAGCTGGAA GACTCCTCCCAG) mouse IL1-β (F:TACAGGCTCCGAGATGAACAAC R: TGCCGTCTTTCATTACACAGGA) mouse IL6 (F: ACTTCCATCCAGTTG

CCTTCTT R: CAGGTCTGTTGGGAGTGGTATC) mouse Sucrase iso-maltase (F: CGTGCAAATGGTGCCGAATA R: TCCTGGCCA TACCTCT CCAA) GAPDH (F: TGACCACAGTCCATGCCATC; R: GACGGACACAT TGGGGGTAG) mouse 45 S (F: GAACGGTGGTGTGTCGTT; R: GCGTCTC GTCTCGTCTCACT). Mouse 18 S: (F: GATGGTAGTCGCCGTGCC; R: CCAAGGAAGGCAGCAGGC).

For splicing validation in mouse intestinal tissues, the MAJIQ package was used to detect differentially spliced genes between *Cbx3* KO and the Ctrl mice. For each sample, counts of reads covering the indicated junctions were used to calculate the proportion of each junction involving the source and the target exons, and represented as histogrammes of percentage of the indicated variant. Splicing index

**Fig. 5 | Non-canonical splicing activity in Ulcerative colitis patients.**
(**a**) Quantification of de novo junctions in healthy donors (Cont, $n = 20$) and in UC patients ($n = 206$) from the PROTECT UC cohort RNA-seq data (GSE109142). De novo junctions were defined as junctions not annotated in the hg19 version of the human genome and not present in both healthy donors and UC patients. Values shown were normalized by the total number of reads in each RNA-seq experiment to account to eventual variations in the depth of the sequencing. *P* Value was calculated with an unpaired two-sided Student's *t* test. Boxplot center indicates the median, the (x) indicates the mean, bounds of the box indicate the 1st and 3rd quartiles, lower and upper whiskers define minima and maxima. When outliers are present, whiskers instead extend 1.5 times the interquartile range. Outliers are shown as dot. (**b**) String chart representing the normalized de novo junction count for each donor (healthy in blue, UC patients in red). Framed areas represent approximately the patients included in the indicated quartiles. (**c**) Genes differentially expressed (>1.5-fold, *P* Val < 0.01) between patients in the lower and the upper quartiles were compared to Gene Ontology annotations. Pathways significantly

($P$ val < 0.01) enriched in the category of "biological processes" are indicated. (**d**) De novo junctions were quantified at genes that, transcriptionally, were affected <1.5-fold between healthy donors and UC patients. For each indicated condition, we counted genes gaining de novo junction 1.5-fold or more ($p$ Val < 0.01). (**e**) Splice junctions in a representative healthy donor and a representative UC patient were visualized with a genome browser in the neighborhood of the *LMNA* gene (indicated in red). Each bar represents a donor/acceptor couple, spanning from the 5′ to the 3′ splice site. The histograms in the top tracks represent the local read density in the RNA-seq data. The UC patient was chosen in the upper quartile but does not display the highest de novo junction count (**f**–**h**) Taqman Assays lamin A and progerin in control ($n = 17$), Crohn disease (CD, $n = 16$) and Ulcerative colitis (UC, $n = 19$) populations, cDNA were extracted from colon biopsies. Mann–Witney U test. (**h**) example of sequencing data from the Taqman PCR end products in one UC patient, Human fibroblast from HGPS patient was used as positive control, UC = Ulcerative colitis. Source data are provided as a Source Data file.

($\Delta\psi$) was calculated using the formula $\Delta\psi = \text{variant1}/(\text{variant1} + \text{variant2})$ and was represented as a ratio or a percentage as indicated in Fig.s. The 6 RNA-seq samples were used to calculate the averages (±deviation) and the $p$ value with the Student's $t$ test (two-sided): $P < 0.05$ (*), $P < 0.01$(**), $P < 0.001$ (***). The primers used for splicing validations are described as Supplementary Table 1 in Supplementary information.

## RNA sequencing and bioinformatic pipelines

Total RNA was extracted from control and *Cbx3* KO epithelial cells from the small intestine (crypts and villi) and colon epithelia ($n = 3$ for each group of mice, so in total $6 \times 3 = 18$ RNA samples) by guanidinium thiocyanate-phenol-chloroform extraction and DNAse treatment, following the manufacturer specifications. Total RNA library preparation and sequencing were performed by Novogene Co., Ltd, as a lncRNA sequencing service, including lncRNA directional library preparation with rRNA depletion (Ribo-Zero Magnetic Kit), quantitation, pooling and PE 150 sequencing (30 G raw data) on Illumina HiSeq 2500 platform. Filtering and trimming of the RNA seq data left around 230–300 million reads pairs/sample.

Gene expression analysis and differential splicing analysis was carried out by Novogene Co., Ltd, using the DESeq2 (v1.18.1) package[67] and rMATS (v4.1.0)[68] (parameters: --libType fr-firststrand −novelSS), respectively, on the mm10 mouse reference genome. For assessment cryptic splice sites, in house RNA-seq data and publicly available Rbm17 KO RNA-seq data (GEO GSE79020) were realigned on the mouse mm9 annotation file from Ensembl (version 67). RNA-seq data from the PROTECT cohort were aligned on the human GRCh37/hg19 annotation file from Ensembl. The resulting BAM files were used to generate bigwig files, using bamCoverage (parameter: --normalizeUsing CPM) from Deeptools (v3.1.3)[69]. Coordinates of annotated and unannotated junctions were then retrieved from the BAM files using regtools (https://github.com/griffithlab/regtools). For each of the 6 conditions, unannotated junctions from the 3 replicates were pooled, before eliminating junctions present both in WT and KO conditions. Finally, junctions spanning over 20 kb were filtered away as possible artefacts of the alignment. The remaining junctions, never shared between WT and mutant samples, were considered de novo. To quantify the strength of the splice sites, DNA sequences comprising the last 3 nucleotides of the exon and the first 6 nucleotides of the intron were recovered for the 5′ end of the junction. For the 3′ end of the junction, the sequences of the last 20 nucleotides of the intron and the first 3 nucleotides of the exon were recovered. Then, the splice sites strength was computed for each junction using the MaxEnt algorithm[69]. The MaxEnt algorithm was also applied to randomized junctions generated by applying the "shuffle" function of the bedtools suite (v.2.25.0)[70] to the bed files containing de novo junctions.

For LTR elements quantification, in house RNA-seq data and HP1 triple KO RNA-seq data (GEO GSE119224) were mapped on the mm10 mouse genome with Ensembl annotation (version 95) using STAR (v2.6.0b)[71] (Parameters: --outFilterMismatchNmax 1 --outSAMmultNmax 1 --outFilterMultimapNmax 30). Read quantification in the different LTR families was carried out with featureCounts (v1.6.1)[72] from the Subread suite (parameters: -s 2 for GSE119244 and parameters: -p -B -C -s 2 for Colon, Crypt and Villi). LTR families (ERK, ERV1, ERVL and ERL-MaLR) were retrieved from the UCSC mm10 RepeatMasker.

GSEA analysis was performed (with -nperm 1000 -permute gene_set -collapse false parameters) using the same expression matrix (v2.2.2), comparing ctrl and *Cbx3* KO samples at the crypt, villi and colon. Transcriptional signatures used for the analysis were extracted from the literature for lgr5 ISC signature[73], enterocyte and Paneth cell signatures[74] and GSEA data bases (MSigDB hallmark gene set).

## Fecal microbiota analysis by 16 S rRNA gene sequencing

Genomic DNA was obtained from fecal samples using the QIAamp power fecal DNA kit (Qiagen), and DNA quantity was determined using a TECAN Fluorometer (Qubit® dsDNA HS Assay Kit, Invitrogen). The V3-V4 hypervariable region of the 16 S rRNA gene was amplified by PCR using the following primers: a forward 43-nuclotide fusion primer 5′ CTT TCC CTA CAC GAC GCT CTT CCG ATC TAC GGR AGG CAG CAG3 consisting of the 28 nt illumina adapter (bold font) and the 14-nt broad range bacterial primer 343 F and a reverse 47-nuclotide fusion 5′GGA GTT CAG ACG TGT GCT CTT CCG ATC TTA CCA GGG TAT CTA ATC CT3′ consisting of the 28 nt illumina adapter (bold font) and the 19-nt broad range bacterial primer 784 R. The PCR reactions were performed using 10 ng of DNA, 0.5 μM primers, 0.2 mM dNTP, and 0.5 U of the DNA-free Taq-polymerase, MolTaq 16 S DNA Polymerase (Molzym). The amplifications were carried out using the following profile: 1 cycle at 94 °C for 60 s, followed by 30 cycles at 94 °C for 60 s, 65 °C for 60 s, 72 °C for 60 s, and finishing with a step at 72 °C for 10 min. The PCR reactions were sent to the @Bridge platform (INRAe, Jouy-en-Josas) for sequencing using Illumina Miseq technology. Single multiplexing was performed using home-made 6 bp index, which were added to R784 during a second PCR with 12 cycles using forward primer (AATGATACGGCGACCACCGAGATCTACACTCTTTCCCTACAC GAC) and reverse primer(CAAGCAGAAGACGGCATACGAGAT-index GTGACTGGAGTTCAGACGTGT). The resulting PCR products were purified and loaded onto the Illumina MiSeq cartridge according to the manufacturer instructions. The quality of the run was checked internally using PhiX, and then, sequences were assigned to its sample with the help of the previously integrated index. High quality filtered reads were further assembled and processed using FROGS pipeline (Find Rapidly OTU with Galaxy Solution) to obtain OTUs and their respective taxonomic assignment thanks to Galaxy instance (https://migale.inra.fr/galaxy). In each dataset, more than 97% of the

paired-end sequences were assembled using at least a 10 bp overlap between the forward and reverse sequences. The following successive steps involved de-noising and clustering of the sequences into OTUs using SWARM, chimera removal using VSEARCh. Then, cluster abundances were filtered at 0.005%. One hundred percent of clusters were affiliated to OTU by using a silva138 16 S reference database and the RDP (Ribosomal Database Project) classifier taxonomic assignment procedure. Richness and diversity indexes of bacterial community, as well as clustering and ordinations, were computed using the Phyloseq package (v 1.19.1) in RStudio software[75]. Divergence in community composition between samples was quantitatively assessed by calculating β-diversity index (UniFrac and weighted UniFrac distance matrices). For the heatmap, a negative binomial model was fit to each OTU, using DESeq2[67] with default parameters, to estimate abundance log-fold changes (FCs). Values of $P$ were corrected for multiple testing using the Benjamini-Hochberg procedure to control the false-discovery rate and significant OTUs were selected based on effect size (FC > |2|, adjusted $P$ value < 0.05).

## Mass spectrometry

**Protein digestion.** Immunoprecipitation eluates were digested following a FASP protocol[76] slightly modified. Briefly, proteins were reduced using 100 mM DTT (dithiothreitol) for 1 h at 60 °C. Proteins were alkylated for 30 min by incubation in the dark at room temperature with 100 μL of 50 mM iodoacetamide. Samples were digested with 2 μL of sequencing grade modified trypsin (Promega, WI, USA) for 16 h at 37 °C. Peptides were collected by centrifugation at 15,000 × $g$ for 10 min followed by one wash with 50 mM ammonium bicarbonate and vacuum dried.

**NanoLC-MS/MS protein identification and quantification.** Peptides were resuspended in 21 μL of 10% ACN, 0.1% TFA in HPLC-grade water prior MS analysis. For each run, 5 μL were injected in a nanoRSLC-Q Exactive PLUS (RSLC Ultimate 3000) (Thermo Scientific,Waltham MA, USA). Peptides were loaded onto a μ-precolumn (Acclaim PepMap 100 C18, cartridge, 300 μm i.d.×5 mm, 5 μm) (Thermo Scientific), and were separated on a 50 cm reversed-phase liquid chromatographic column (0.075 mm ID, Acclaim PepMap 100, C18, 2 μm) (Thermo Scientific). Chromatography solvents were (A) 0.1% formic acid in water, and (B) 80% acetonitrile, 0.08% formic acid. Peptides were eluted from the column with the following gradient 5–40% B (38 min), 40–80% (1 min). At 39 min, the gradient stayed at 80% for 4 min and, at 43 min, it returned to 5% to re-equilibrate the column for 16 min before the next injection. Two blanks were run between each series to prevent sample carryover. Peptides eluting from the column were analyzed by data dependent MS/MS, using top-10 acquisition method. Peptides were fragmented using higher-energy collisional dissociation (HCD). Briefly, the instrument settings were as follows: resolution was set to 70,000 for MS scans and 17,500 for the data dependent MS/MS scans in order to increase speed. The MS AGC target was set to $3.10^6$ counts with maximum injection time set to 200 ms, while MS/MS AGC target was set to $1.10^5$ with maximum injection time set to 120 ms. The MS scan range was from 400 to 2000 m/z.

**Data analysis following nanoLC-MS/MS acquisition.** Raw files corresponding to the proteins immunoprecipitated were analyzed using MaxQuant 1.5.5.1 software against the Human Uniprot KB/Swiss-Prot database 2016-01[77]. To search parent mass and fragment ions, we set a mass deviation of 3 ppm and 20 ppm respectively, no match between runs allowed. Carbamidomethylation (Cys) was set as fixed modification, whereas oxidation (Met) and N-term acetylation were set as variable modifications. The false discovery rates (FDRs) at the protein and peptide level were set to 1%. Scores were calculated using Max-Quant as described previously[77]. Peptides were quantified according to the MaxQuant MS1 signal intensities. Statistical and bioinformatic

analysis, including volcano plot, were performed with Perseus software version 1.6.7.0 (freely available at www.perseus-framework.org). For statistical comparison, we set two groups, IP and negative control, each containing 3 biological replicates. We then retained only proteins that were quantified 3 times in at least one group. Next, the data were imputed to fill missing data points by creating a Gaussian distribution of random numbers with a standard deviation of 33% relative to the standard deviation of the measured values and 3 standard deviation downshift of the mean to simulate the distribution of low signal values. We performed a $T$-test, and represented the data on a volcano plot (FDR < 0.05, S0 = 1).

## Protein extraction, Co-immunoprecipitation, SDS-PAGE and immunoblotting

HEK293T (CRL-1573, ATCC) or HeLa cells (CCL-2, ATCC) were extracted in TNEN300:0.1 buffer (50 mM Tris pH 8, 300 mM NaCl, 0.1% NP40, 1 U/μl RNasin (Promega), 1X protease inhibitor (Roche)) for 30 min on ice with resuspension through syringe gauge. The denatured chromatin was removed by pipetting out. Extracts were diluted to 150 mM NaCl and half was supplemented by 0.2 μg/mL RNaseA (DNase free, Roche) or by 0.1 U/μL RNasin. One fourth of the 10 cm dish was immunoprecipitated overnight at 4 °C on wheel by 3 μg of anti-V5 antibodies for transfected HEK293 cells and by 1.5 μg of anti-SRSF1 antibodies for HeLa cells. Complexes were recovered by G-protein dynabeads (Dynal) for 2 h at 4 °C on wheel. Beads were washed 5 times in 1 mL of TNEN150:0.1 during 7 min at 4 °C on wheel before denaturation in loading buffer at 100 °C during 10 min. Proteins were resolved on SDS-PAGE Criterion 4–12% (Bio-Rad).

Intestinal epithelial cells purified from mice (at least $n$ = 3 separate experiments) were lysed at 4 °C in a buffer containing 25 mM Tris pH 7.5, 1 mM EDTA, 0.1 mM EGTA, 5 mM MgCl₂, 1% NP-40, 10% Glycerol, 150 mM NaCl, and then cleared by centrifugation at 14,000 $rpm$ for 30 min at 4 °C. Proteins were separated on SDS−PAGE gels and transferred to nitrocellulose membranes by standard procedures. The following antibodies have been used: mouse anti-progerin monoclonal antibody (13A4DA, sc-81611, Santa Cruz, dilution 1/50), anti-HP1α (2H4E9, Novus Biologicals, dilution 1/100), HP1β (1MOD-1A9, Thermo Scientific, dilution 1/100) and HP1γ (2MOD-IG6, Thermo Scientific, dilution 1/100) and γ Tubulin (4D11, Thermo Scientific, dilution 1/2000) antibodies. anti-HA (12CA5, Sigma, Dilution 1/1000), anti-Flag M2 (Sigma, dilution 1/1000), anti-V5 (Bethyl A190-120A, dilution 1/1000), anti-SRSF1 (Santa Cruz sc-33652, dilution 1/250) antibodies, anti-Rabbit IgG Bright700 (BioRad, dilution 1/5000), anti-Mouse IgG Bright700 (BioRad, dilution 1/5000). Western blot signal was acquired with Chemidoc Imaging (Bio Rad).

## Statistical analyses

Statistical analyses were performed with the GraphPad Prism software. Differences between groups were assayed using a two-sided Student's $t$ test using GraphPad Prism. In all cases, the experimental data were assumed to fulfill $t$-test requirements (normal distribution and similar variance); in those cases, where the assumption of the $t$-test was not valid, a nonparametric statistical method was used (Mann−Whitney test). The differences between three or more groups were tested by one-way ANOVA. A $p$ value < 0.05 was considered as significant. Error bars indicate the standard error of the mean.

## Reporting summary

Further information on research design is available in the Nature Portfolio Reporting Summary linked to this article.

# Data availability

RNA-seq data, the derived mm9.bigwig files, and the de novo junction.bed files generated in this study have been deposited at NCBI Gene Expression Omnibus database and are accessible through GEO

Series accession number GSE192800. Proteomics data have been deposited to the ProteomeXchange Consortium via the PRIDE partner repository with the dataset identifier PXD031580. The 16 S rRNA gene sequencing data generated in this study care are available via BioProject accession numbers PRJNA803986.

RNA-seq raw fastq files from GSE109142 (UC PROTECT cohort), GSE145309 (mouse ES Crispr KO Zc3h13) and GSE119244 (triple knockout of HP1α, HP1β, and HP1γ mouse liver) were obtained from NCBI's GEO database. PRJNA669300 (Ppil1A99T/A99T KI E14.5 brains) and PRJNA708182 (SRSF5 KO mouse heart) were obtained from BioProject's database. Source data are provided with this paper.

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

## Acknowledgements

We thank Florence Cammas for providing the *Cbx3*^fl/fl^ mice. We thank Dr. Zhou Zhongjun (Department of Biochemistry, Li Ka Shing Faculty of Medicine The university of Hong Kong) for providing us MEF from HGPS mice. We thank Dr. Cohen-Tannoudji for providing the Villin-CreERT2 mice (Institut Pasteur, Paris, France). This work has been supported by the «Agence National de la Recherche» (ANR) grant (EPI-CURE, R16154KK by L.A.) and REVIVE (by C.M.), an ANR 'Laboratoire d'Excellence' program (2011–2021).

## Author contributions

J.M-G. designed and performed most of the experiments, and participated in the writing of the paper; Y.X. performed experiments on splicing and Y.C-M. on HGPS mice; C.R. performed QPCR experiments on mice *Cbx3* KO and IBD patients; E.A. performed immunofluorescence studies; I.C.G. performed proteomic analysis; I.C. produced EM data, A.B. performed DNA extraction and PCR for microbiota analysis; C.C. performed microbiota analysis from 16 S sequencing; X.T., E.O.D. provided medical reports and clinical samples and A.D. provided patients cDNA and QPCR data; E.B. designed, carried out, and interpreted experiments for splicing, ddPCR and participated in the writing of the paper; M.C. carried out most bioinformatics on the RNA-seq data from the mouse model and from the UC patients; G.R. and M.E. provided HGPS mice; C.M. designed and participated in the bioinformatics exploring splicing, and in the writing of the paper. L.A. conceived the project, supervised the study, wrote and edited the paper.

## Competing interests

The authors declare no competing interests.
