## [Peer Review File · Nature Communications]

The Heterochromatin protein 1 is a regulator in RNA splicing precision deficient in ulcerative colitisREVIEWER COMMENTS

Reviewer #1 (Remarks to the Author):

The manuscript by Mata-Garrido and colleagues described a well-designed study that investigate the role of HP1gamma in ulcerative colitis (UC). Intriguingly, the authors found that HP1gamma is down-regulated in UC, which leads to an increase of a wide-spread splicing "noise" (cryptic splicing). One of the mis-regulated splicing events is the increased production of the progerin mRNA transcript of the Lamin A gene.

The study was well controlled and executed. The conclusions are solid. This reviewer has the following comments to help the authors improve their manuscript.

1. It would be nice to see more RNA data in addition to western blot and immunostaining analysis. For example, RT/PCR analysis of the Lamin A transcripts would strengthen the arguments regarding the Progerin production.
2. Is it possible to re-introduce HPgamma in the cells where HPgamma was decreased such as in Cbx3 KO mice?
3. In Figure 3B, are there really 200K de novo junctions in control cells? This number is really high. How was de novo junctions defined? Is this consistent with previous studies on splicing noise?
4. Minor points: on line 206, it should be "de novo"; on line 234, it should be aging.

Reviewer #2 (Remarks to the Author):

This manuscript by Mata-Garrido and colleagues describes the role of HP1g in ulcerative colitis in both humans and mice. This link in the human condition is first derived from the low expression of HP1g in diseased tissues. The targeted depletion of HP1g in mice intestinal tissues reproduces many of the disease features of ulcerative colitis and is associated with a change in the microbiome known to occur in inflammatory bowel diseases. Overall, despite the lack of details on molecular mechanisms, the results presented are interesting, and the link with progerin production, and therefore aging, is compelling.

1. In terms of mechanisms, the manuscript would benefit from addressing at least some of the following questions:
 - a. Why is HP1g expression down in UC?
 - b. How can a change in HP1g lead to a change in the microbiome?
 - c. Most importantly and perhaps more relevant to the current work: How does Δ HP1g lead to splicing noise? The fact that HP1g Interacts with spliceosome factors (but apparently not with RBM17, TP43 (TDP43?) and hnRNP proteins that have been implicated in splicing noise) is not informative as far as mechanisms are concerned. There are very few studies that have examined splicing noise following the depletion of known regulators or spliceosome components. Splicing noise may be a feature associated with any type of interference with splicing activity. Moreover, suboptimal spliceosomal activity may not only produced more cryptic splice site usage but also induce more exon skipping events. Also, could an uncoupling from transcription lead to more aberrant splicing events? The absence of hints for mechanisms makes the discussion somewhat rambling.
2. Fig. 1 and elsewhere. Please indicate what are the vertical lines on the x-axis.
3. IL1 and IL6 expression are also indicative of senescence (SASP factors). Given the later link with aging (progerin), the expression status of other senescence markers (b-gal, p21) would be informative. What about expression of HP1g in senescent cells?
4. Were RNases included in the samples used to identify HP1g interactants by mass spectrometry. This is important because changes could be caused by RNA bridging and lead to differences if the overall

RNA abundance is different in different conditions.

5. In Fig. 3d, it is unclear if the increase in de novo events is caused by less RNA in cells, allowing sequencing to sample more rare events for a similar absolute number of reads.

6. Fig. 3e. It is unclear what the bars represent (bars are mostly of similar sizes, any reason for that?).

7. It is intriguing that both more lamin A and more progerin splice products are detected in mutant mice (lane 247). If the ratio of lamin A/progerin is the same in normal and mouse tissues, is it just that the lamin transcripts are more abundant? (in contrast to what the text indicates on page 15, line 248). This is in contrast with Fig4 i/j, where progerin products are more abundant only in human UC tissues. Since both laminA and progerin mRNA seemingly increase compared to the control, it would be more informative to show a PSI (percentage splicing index) analysis for all tissues and progeria cells using endpoint PCR with primers flanking the event or alternatively, ddPCR using the qPCR primers.

8. Page 4. Line 108. "OTU" not defined.

9. Figure 2f. The difference between ctrl and KO is unclear as both show expression of nucleolin at the basis of villi, and not in the upper part.

10. Page 15. Line 264-271. The results presented here do not support that progerin is sufficient to induce toxicity, but only that toxicity is induced by the cbx3 KO. The first sentence should be rephrased.

11. Figure 4d. It is not clear which band is progerin. Can we assume progerin is the top band? and why is the lower band not seen in Extended figure 7?

Reviewer #3 (Remarks to the Author):

In this study, Mata-Garrido, et al. show that HP1 γ expression is reduced in both ulcerative colitis patients and Il10/Nox1 DKO mice. Deletion of HP1 γ affected the gene expression of metabolism and inflammation in gut epithelial cells, and altered composition of the gut microbiota. Genetic depletion of HP1 γ increased BrdU/EdU positive cells, OLFM4 positive cells, and stem cell signature intestinal crypt in vivo and organoids in vitro, while Paneth cells were reduced. Next, the authors focus on the role of HP1 γ on pre-mRNA splicing based on their mass spectrometry findings of incorporation of HP1 γ in spliceosome, and deletion of HP1 γ exerts significant impacts on splicing process which has been analyzed by rMATS. Lastly, lamin A is picked out as a representative gene with altered composition of splice variants. Progeria gene is increased in the patients with HGPS and UC. Although some topics are of interest, but the whole structure is not logically connected. In addition, clinical evidence of co-occurrence of HGPS and UC has not been confirmed.

Lmna is affected by the loss of HP1 γ in the villi, but it is listed as one of the 84th genes with highest Mut/WT ratio. This is a quite biased approach and the 83 genes with higher Mut/WT ratio should not be neglected. In addition, Lmna is the least affected genes in the colon that are significantly different between Mut vs WT. Therefore, this reviewer cannot figure out why alternative splicing of Lmna explains alteration of the crypt bottom in the small bowel and colon and pathogenesis of UC or animal models of colitis.

As the author mentions in the manuscript, Lmna splicing was not significantly different between HP1 γ sufficient and deficient crypts, but rMATS detects many new variants.

rMATS analysis has revealed far more variations in Lmna than previously recognized. The authors should recognize the possibility that rMATS are vulnerable to the outliers when dealing with pooled samples and show

HP1 γ recognizes methylation of H3K9, or other histones, and is important in heterochromatin formation. A more comprehensive investigation of the effects of HP1 γ on global H3K9 methylation.

Figure 2 shows increase in BrdU positive cells in the crypt bottom of Cbx3 KO mice or intestinal organoids, but the authors should clarify BrdU or EdU positive cells overlap stem cells. In addition, Lgr5 is a key marker besides Ascl2 and Olfm4, and this helps the authors' hypothesis that most of the HP1 γ -deleted intestinal stem cells are in S phase.

Minor points:

This reviewer wonders why the authors do not clearly state the gene names encoding HP1 γ and lamin A. This information will help broader readers of Nature Communications.

Table legends are missing in the PDF file.

There are some typos (l1139; Edu, l1289;potentiel).

laminA and lamin A should be unified.

Table 9 show exactly the same number of variants, such as Ugt1a7c, Ugt1a8, and Ugt1a9. This reviewers suspect this software cannot recognize the small differences of the sequence in the gene.

Response to Reviewer's comments

Reviewer #1 (Remarks to the Author):

The manuscript by Mata-Garrido and colleagues described a well-designed study that investigate the role of HP1 γ in ulcerative colitis (UC). Intriguingly, the authors found that HP1 γ is down-regulated in UC, which leads to an increase of a wide-spread splicing "noise" (cryptic splicing). One of the mis-regulated splicing events is the increased production of the progerin mRNA transcript of the Lamin A gene.

The study was well controlled and executed. The conclusions are solid. This reviewer has the following comments to help the authors improve their manuscript.

1. It would be nice to see more RNA data in addition to western blot and immunostaining analysis. For example, RT/PCR analysis of the Lamin A transcripts would strengthen the arguments regarding the Progerin production.

We now provide end-point RT-PCR data using primers nested on exon 9 and 12, for detection of either laminA or progerin transcripts (primer choice according to *Osorio et al.*¹). These data evidence for the production of progerin transcript in Cbx3 KO, but not in WT mice (intestinal epithelia from HGPS G609G mice were used as positive controls). These data are now provided as a new **Extended data Figure 8a**.

2. Is it possible to re-introduce HP γ in the cells where HP γ was decreased such as in Cbx3 KO mice?

While we agree that re-introducing HP1 γ would allow to further document that our phenotype is solely due to *Cbx3* inactivation, such a rescue experiment, that would involve the engineering of an addition transgenic mouse model, would unfortunately not be feasible in the time allowed for the revision of the manuscript.

3. In Figure 3B, are there really 200K de novo junctions in control cells? This number is really high. How was de novo junctions defined? Is this consistent with previous studies on splicing noise?

To produce the figures in Panel 3B, the triplicated RNA-seq data from each condition were first pooled (3 villi WT, 3 villi Cbx3 KO, 3 crypts WT...). Then, the RegTools package was used to identify split reads in each pool, and to compare these reads with an annotation file to determine whether they covered previously described junctions or "de novo" junctions. We finally filtered out "de novo" junctions detected in both WT and in Cbx3 KO RNA-seq data. The output of this pipeline is indeed in the range 200K "de novo" junctions in the WT, which represent in average 8-10 unannotated junctions per gene, mostly originating from small variations in the exact splice site or from unannotated exon skipping events. As Panel 3b also includes a re-analysis of an RNA-seq experiment documenting the impact of Rbm17 KO on splicing noise², we hope the Reviewer will agree that abundance of de novo junctions in our data is in a range comparable to that observed in earlier studies.

4. Minor points: on line 206, it should be "de novo"; on line 234, it should be aging.

These modifications have been provided

Reviewer #2 (Remarks to the Author):

This manuscript by Mata-Garrido and colleagues describes the role of HP1 γ in ulcerative colitis in both humans and mice. This link in the human condition is first derived from the low expression of HP1 γ in diseased tissues. The targeted depletion of HP1 γ in mice intestinal tissues reproduces many of the disease features of ulcerative colitis and is associated with a change in the microbiome known to occur in inflammatory bowel diseases. Overall, despite the lack of details on molecular mechanisms, the results presented are interesting, and the link with progerin production, and therefore aging, is compelling.

1. In terms of mechanisms, the manuscript would benefit from addressing at least some of the following questions:

a. Why is HP1 γ expression down in UC?

At this stage, we have documented that the decreased HP1 γ expression associated with UC is not transcriptional. RT-qPCR analysis of mRNA from our cohort of IBD patients and healthy controls yielded no evidence for a down-regulation of HP1 γ mRNA expression in IBD colon tissues (information provided in **Supplementary data table 14**). Instead, we are currently investigating a possible impact of ER stress on HP1 γ protein accumulation. Indeed, we observed reduced levels of HP1 γ protein in the EXCY2 mouse, a model relevant for UC in which the pan-colitis in the colon epithelium is linked to early and chronic ER stress³. In addition, using human enterocytic cell lines, we found that ER stressors such as thapsigargin affect HP1 γ protein expression. These data are provided below for the Reviewer. However, the investigation on this issue being still at an early stage, we would prefer not to include them in the manuscript.

Legend: TC7 and HT29-MTX enterocytic cell lines were incubated with thapsigargin at the indicated doses during 16h or vehicle (DMSO). (a-b) TC7 cells: (a) immunoblot with anti-HP1 γ antibody in lysates which appears as a doublet (arrow) in this cell line (b) quantification of the HP1 γ signal as related to β -Actin by Image J (c-d) HT29-MTX cells: (c) immunoblot with anti-HP1 γ antibody in lysates and (d) quantification of the HP1 γ signal as related to γ -tubulin by Image J (representative of n=3 separate experiments/cell line).

b. How can a change in HP1 γ lead to a change in the microbiome?

Reduced levels of HP1 γ may affect the microbiome in multiple manners. Firstly, we showed that HP1 inactivation led to an increase in inflammatory gene expression, and inflammation is in itself conducive to a blooming of *Enterobacteriaceae*, at the expense of symbiotic bacteria. Moreover, the transcriptomic analysis at both small

intestine and colon epithelia showed that HP1 γ deficiency profoundly affected mRNA expression of multiple antimicrobial peptides (AMPs), whose antimicrobial activities play essential functions in the homeostatic shaping of the microbiota (for review,⁴). In particular, in the *Cbx3* KO mouse epithelia, compromised mRNA expression were detected for defensins, cathelicidins, and regenerating gene (Reg) type of AMPs. This has now been confirmed by RT-QPCR at villi and colon epithelia. These data have been integrated in the revised version of the manuscript as **new Supplementary Table 3** and the RT-QPCR data as **new Figure 1e**.

c. Most importantly and perhaps more relevant to the current work: How does Δ HP1 γ lead to splicing noise? The fact that HP1 γ Interacts with spliceosome factors (but apparently not with RBM17, TP43 (TDP43?) and hnRNP proteins that have been implicated in splicing noise) is not informative as far as mechanisms are concerned. There are very few studies that have examined splicing noise following the depletion of known regulators or spliceosome components. Splicing noise may be a feature associated with any type of interference with splicing activity. Moreover, suboptimal spliceosomal activity may not only produced more cryptic splice site usage but also induce more exon skipping events. Also, could an uncoupling from transcription lead to more aberrant splicing events? The absence of hints for mechanisms makes the discussion somewhat rambling.

By data mining, we have now identified several molecular partners of HP1 γ , that when inactivated, result in a significant increase in the splicing noise. Notably, the Ser/Arg-rich SRSF5⁵, previously shown to reduce progerin by redirecting splicing towards lamin A 5' splice site utilization^{6,7}, PPL1, a peptidyl prolyl isomerase-like spliceosome component whose brain inactivation leads to splicing alterations with aberrant use of weak splicing sites⁸ and ZC3H13, a zinc-finger protein regulating RNA modification at N6-methyladenosine⁹, a modification involved in various aspects of RNA metabolism, including RNA splicing¹⁰⁻¹². The data in the revised manuscript are shown in **Extended data Figures 7a-c**, and lead us to suggest that HP1 γ reduces splicing noise by favoring recruitment to the chromatin template of proteins involved in the precision of splicing decisions.

Regarding exon-skipping events being a component of noise, we fully agree with the Reviewer, and exon-skipping events are accounted for in our evaluation of splicing noise. Indeed, to produce the figures in Panel 3B, we used the RegTools package to identify split reads mapping outside annotated junctions. This has now been made clear in the text.

Finally, regarding a possible uncoupling between transcription and splicing in the absence of HP1, and the impact of this uncoupling on splicing noise, we do not have at this point have any data on the issue, but we now mention this possibility in the discussion.

2. Fig. 1 and elsewhere. Please indicate what are the vertical lines on the x-axis.

It is not clear to us what vertical lines the Reviewer is referring to. Is it possible that Figures were degraded during PDF conversion or downloading?

3. IL1 and IL6 expression are also indicative of senescence (SASP factors). Given the later link with aging (progerin), the expression status of other senescence markers (b-gal, p21) would be informative. What about expression of HP1 γ in senescent cells?

In vivo, *Cbx3* inactivation was associated with increase detection of proliferative

marks and cell maturation defects (**Figures 2a-c, Extended data Figures 4-5**), while the RNA-seq data indicated unchanged expression of p21/Cdkn1a and p16/Cdkn2a. We therefore do not suspect *in vivo* HP1 γ inactivation to be associated with senescence. Moreover, apart from senescence, several biological functions have been attributed to progerin such as alterations of cell fate and differentiation, modulation of energy metabolism as well as nucleolar expansion and translation¹³⁻¹⁶. The potential impact of progerin on these functions will be part of a future investigation.

Mining of a large set of RNA-seq data from different cell types (melanocytes, keratinocytes, and fibroblasts) driven into senescence by ionizing radiations¹⁷ showed only a moderate (less than 2-fold) decrease in *Cbx3* mRNA levels upon exit of the cells from the proliferative state (figure 1a below). Moreover, our in-house Western blots on IMR90 or WI38 cells driven into senescence by oncogenic Raf showed that levels of HP1 were slightly increased (figure 1b below). These data are provided below for the Reviewer and not included in the manuscript.

Legend: in (a) Indicated primary cells were exposed to ionizing radiation and then kept in culture for up to 20 days. Senescence was typically detected approximately 10 days after radiation. Cbx3 transcription drops between 30 and 40% at early phases of the experiments, possibly as a consequence of the DNA damage or the growth arrest, then remains essentially stable throughout the remaining time points. These observations strongly suggest that cellular senescence is not correlated with reduced expression of the Cbx3 gene. Data set extracted from ID code ArrayExpress: E-MTAB-5403 in (b) Total protein extracts were prepared from IMR90 or WI38 ER-RasV12 human fibroblasts either proliferating (Ctrl) or driven into senescence by induction of an activated form of RAS (Tamox). Expression of the indicated proteins was assessed by immunoblot (IB). Senescence was induced with 20 nM of 4-hydroxy-tamoxifen during 3 days for the expression of C-RAF and RASval12 in IMR90 cells. Senescence-associated β -galactosidase assays were performed by staining formaldehyde (4%)-fixed cells with 1 mg/ml 5-bromo-4-chloro-3-indolyl- β -D-galactoside (X-Gal) in 40 mM citric acid-sodium phosphate (pH 6.0), 5 mM potassium ferrocyanide/5 mM potassium ferricyanide, 150 mM NaCl and 2 mM MgCl₂ buffer 16 h at 37°C. Images were acquired on an Olympus microscope at 20 \times magnification

4. Were RNases included in the samples used to identify HP1 γ interactants by mass spectrometry. This is important because changes could be caused by RNA bridging and lead to differences if the overall RNA abundance is different in different conditions.

To address this issue, we tested the impact of RNase on co-immunoprecipitation of HP1 γ with the splicing factors Tra2b/SFRS10 (identified in our interactome), SRSF3/SRp20, RBM39/CAPER α , and SRSF1 for which antibodies were available.

The interactions of HP1 γ with TRA2b, SRSF1 and in a less extent with RBM39 were not affected by the degradation of the RNA present in the extracts (**Extended data Figure 6c-d, revised version**). Interaction of HP1 γ with component of the spliceosome has been documented in previous publications¹⁸⁻¹⁹ and in a previous work, we showed that the molecular complex associating Ago proteins, HP1 γ , and components of the spliceosome was largely resistant to treatment with RNases, ruling out a possible aggregation through a RNA scaffold²⁰.

5. In Fig. 3d, it is unclear if the increase in de novo events is caused by less RNA in cells, allowing sequencing to sample more rare events for a similar absolute number of reads.

While the depth of the RNA-seq has a great impact on the detection of rare splicing events, eventual differences in the RNA contained per cell in the different samples is leveled out during the making of the libraries. At that stage, RNA preps are carefully quantified and equal amounts are used for each reaction.

6. Fig. 3e. It is unclear what the bars represent (bars are mostly of similar sizes, any reason for that?).

Each bar represents a “de novo” donor/acceptor couple. They span from the 5’ to the 3’ splice site (thereby representing the “intron” getting spliced out). This is now indicated in the legend of Figure 3. Some of these bars symbolize splicing events differing only by a few nucleotides, which at the given scale may leave the impression that they have the same length. Yet, all the bars are different.

7. It is intriguing that both more lamin A and more progerin splice products are detected in mutant mice (lane 247). If the ratio of lamin A/progerin is the same in normal and mouse tissues, is it just that the lamin transcripts are more abundant? (in contrast to what the text indicates on page 15, line 248). This is in contrast with Fig4 i/j, where progerin products are more abundant only in human UC tissues. Since both laminA and progerin mRNA seemingly increase compared to the control, it would be more informative to show a PSI (percentage splicing index) analysis for all tissues and progeria cells using endpoint PCR with primers flanking the event or alternatively, ddPCR using the qPCR primers.

We thank the Reviewer for these suggestions that we believe have really helped documenting the impact of HP1 γ inactivation on the production of progerin. We have now performed endpoint semi-quantitative RT-PCR using primers nesting on exon 9 and 12 for detection of lamin A and progerin transcripts (primers choice according to Osorio *et al*,¹ and intestinal crypt epithelium from HGPS G609G mice used as positive controls). Endpoint RT-PCR showed evidence for progerin transcript detection in *Cbx3* KO but not in Ctrl mice (**Extended data Figure 8a**).

Extended data Figures 8b-c showed the data obtained with Taqman quantitative PCR in crypt and villi (related to GAPDH, **Extended Figure 8b**) and in those samples, quantitative measurement of progerin and lamin A transcripts by ddPCR (**Extended Figure 8c**) to further determine the Percent Spliced In (PSI) score of the progerin-specific splicing event (**Extended Figure 8d**). For ddPCR calibration and comparison with HGPS, cDNA from purified crypt epithelial cells of HGPS mice was also used. In Ctrl crypts and villi, we detected an average of 415 \pm 166 copies/ul and 0.16 \pm 0.15 copies/ul of canonical and progerin-specific splicing products respectively, while, in the *Cbx3* KO, these figures raised to 1566 \pm 725 copies/ul for Lamin A and 2.8 \pm 0.42 copies/ul for progerin (**Extended Figure 8c**). As expected,

progerin detection in cDNA derived from purified crypt epithelial cells of HGPS mice was higher with 33.86 copies/ul and 11,8 copies/ul of lamin A. Translation into average PSIs showed an increase from 0.045 to 0.21 upon inactivation of *Cbx3*, documenting an increased occurrence of the progerin-specific splicing event that was not due to the increased expression of the *Lmna* gene in the mutant mice (**Extended Figure 8d** and **Supplementary data Table 11**). These data also confirmed that in comparison with progeria epithelial cells, usage of the aberrant progerin splice site induced by *Cbx3* inactivation remained less frequent, in coherence with the increased strength of the progerin 5'SS provided by the HGPS mutation⁷.

8. Page 4. Line 108. "OTU" not defined.

OTU for Operational Taxonomic Unit is now mentioned.

9. Figure 2f. The difference between ctrl and KO is unclear as both show expression of nucleolin at the basis of villi, and not in the upper part.

To improve the readability of the Panel, we have now provided an insert with higher magnification to the regions of interest (**Figure 2f**).

10. Page 15. Line 264-271. The results presented here do not support that progerin is sufficient to induce toxicity, but only that toxicity is induced by the *cbx3* KO. The first sentence should be rephrased.

The sentence has been rephrased as follows: Accumulation of progerin chiefly disrupts the structure of the nuclear lamina. We thus examined whether toxicity to the nuclear lamina was detected upon *Cbx3* inactivation.

11. Figure 4d. It is not clear which band is progerin. Can we assume progerin is the top band? and why is the lower band not seen in Extended figure 7?

Progerin is indeed the top species, only detected KO *Cbx3* mice lysates, in consistence with the RNA data, while the bottom band seen in both Ctrl mice and KO mice is non-specific. We are highly confident that we are detecting progerin in this blot, as we used the well-characterized monoclonal anti-progerin 13A4DA antibody, extensively used by the community for progerin detection by immunofluorescence and immunoblot (PMID: # 28674081 ; PMID: # 27739443, PMID: # 33907225). The nonspecific species also appears under the conditions used in **Extended figure 7 (Extended figure 9 in revised manuscript)**, but remains invisible due to the shorter exposure times required for the very abundant progerin signal obtained with colon extracts from HGPS G609G mice. In **Extended figure 9**, we provided the 8-16% gradient gel image allowing to visualize the non-specific bottom species, and for Reviewer the image of the whole blot.

Reviewer #3 (Remarks to the Author):

In this study, Mata-Garrido, et al. show that HP1 γ expression is reduced in both ulcerative colitis patients and I110/Nox1 DKO mice. Deletion of HP1 γ affected the gene expression of metabolism and inflammation in gut epithelial cells, and altered composition of the gut microbiota. Genetic depletion of HP1 γ increased BrdU/EdU positive cells, OLFM4 positive cells, and stem cell signature intestinal crypt in vivo and organoids in vitro, while Paneth cells were reduced. Next, the authors focus on the role of HP1 γ on pre-mRNA splicing based on their mass spectrometry findings of incorporation of HP1 γ in spliceosome, and deletion of HP1 γ exerts significant impacts on splicing process which has been analyzed by rMATS. Lastly, lamin A is picked out as a representative gene with altered composition of splice variants. Progeria gene is increased in the patients with HGPS and UC. Although some topics are of interest, but the whole structure is not logically connected.

We thank the reviewer for his interest in our observations, and we hope that the new version that now contains several additional patient data and validation experiments will appear better structured.

In addition, clinical evidence of co-occurrence of HGPS and UC has not been confirmed.

We here need to stress that we do not claim that there is a co-morbidity between UC and HGPS. Rather, splicing alterations characterize these 2 diseases. More specifically, while HGPS patients produce an alternative spliced version of lamin A transcript (progerin) due to a genomic mutation, we propose that UC patients produce the same lamin A splice variant as a result of increased splicing noise.

In the first version of the manuscript, we demonstrated the increased production of progerin in our cohort of UC patients (now integrated in **Figure 5 f-h**, revised manuscript). In the new version, we additionally document that UC is associated with an extensive increase in splicing noise (**Figure 5a-e**, and **Supplementary data Tables 12 and 13**). We analyzed 206 UC transcriptomes from the largest prospective multicenter inception treatment-naïve UC cohort PROTECT, whose core transcriptional signature has been validated across several independent pediatric and adult UC cohorts^{21,22}. This data set was also selected because it had a depth of sequencing sufficient for splicing analysis. We showed a potent increase use of non-canonical splice sites in the group of UC patients (**Figure 5a**, 20 healthy controls and 206 UC patients, pVal in the range of 10^{-30}), affecting the splicing of multiple genes including LMNA (**Figures 5d-e**, **Supplementary data Table 13**). Our transcriptome analysis also shows that UC patients with highest splice disturbance exhibit a gene signature previously reported in severe UC²¹ (i.e enrichment in inflammatory genes and a substantial down-regulation in mitochondrial genes, **Figure 5b-c and Supplementary data Table 12**). Consistent with this, we found that splice noise correlates with a more severe histologic severity score (p-Val=0.03) and fewer clinical remission at 4 weeks (**Extended data Figure 12 a-b**).

Thus, from the data obtained in these 2 IBDs cohorts, we propose a model in which disturbance in RNA splicing precision in UC increases detection of rare transcripts variants, exemplified by progerin.

Lmna is affected by the loss of HP1 γ in the villi, but it is listed as one of the 84th genes with highest Mut/WT ratio. This is a quite biased approach and the 83 genes with higher Mut/WT ratio should not neglected. In addition, Lmna is the least affected

genes in the colon that are significantly different between Mut vs WT. Therefore, this reviewer cannot figure out why alternative splicing of Lmna explains alteration of the crypt bottom in the small bowel and colon and pathogenesis of UC or animal models of colitis.

We certainly do not suggest that LMNA alterations recapitulates the pathogenesis of UC, and we expect many other alternatively spliced proteins be involved in the disease, as suggested previously^{23,24}. This has now been made clearer in the discussion as followed: “the increased detection of progerin in UC colon tissue prompts us to speculate that, alike what we observed in the *Cbx3* KO mice, the lamin A splice variant is only the “tip of the iceberg”, indicative of a more extensive disturbance in RNA splicing precision endured by UC patients, and clearly illustrated in the PROTECT patient cohort”. Regarding the focus on LMNA, it is one of the few genes for which an AS product is known to interfere with the biology of the cell. We therefore hope that the Reviewer will agree that an in-depth examination of this particular gene would be of interest in the context of UC.

As the author mentions in the manuscript, Lmna splicing was not significantly different between HP1 γ sufficient and deficient crypts, but rMATS detects many new variants. rMATS analysis has revealed far more variations in Lmna than previously recognized. The authors should recognize the possibility that rMATS are vulnerable to the outliers when dealing with pooled samples and show

We realize that we have been misleading on our usage of rMATS and in our description of Figure 3E. The rMATS software was only used to gain a global insight on variations in the usage of annotated junctions upon inactivation of *Cbx3* (**Figure 3a**). Analysis of the splicing noise was carried out by counting split reads not matching annotated junctions. Unlike a regular “splicing analysis” focusing on relative abundance of specific splicing events, the objective of our approach was to document that *Cbx3* inactivation results in spurious splicing reactions. With this approach, we establish that, upon *Cbx3* inactivation, the total number of non-annotated splicing events is significantly increased. **Figure 3e** illustrates this significantly increased splicing noise at 3 genes relevant for UC. This has been mentioned in the text and in the caption of Figure 3.

HP1 γ recognizes methylation of H3K9, or other histones, and is important in heterochromatin formation. A more comprehensive investigation of the effects of HP1 γ on global H3K9 methylation.

The global level of H3K9me3 in epithelial is not impacted by *Cbx3* inactivation in the small intestine and in the colon. This is now documented by immunofluorescence experiments followed by ImageJ quantification of the signal (**Extended Figure 1b-d**).

Figure 2 shows increase in BrdU positive cells in the crypt bottom of Cbx3 KO mice or intestinal organoids, but the authors should clarify BrdU or EdU positive cells overlap stem cells. In addition, Lgr5 is a key marker besides Ascl2 and Olfm4, and this helps the authors' hypothesis that most of the HP1 γ -deleted intestinal stem cells are in S phase.

Position of stem cells in the crypts is well-established from position 0 to +4 and **Figure 2b** showed a quantification of BrdU positive cells at these positions. The

signal at these positions can thus be unambiguously attributed to stem cells. This has now been made clearer in the text.

In the ex-vivo organoid model, the stem cells localize to the buds, but the changes induced by *Cbx3* inactivation were mainly observed in villi domain, as shown in **Figure 2c** and the quantification in crypt and villi is provided in **Extended data Figure 5c**.

Regarding the usage of *Lgr5* as a marker of stem cells, there is unfortunately no good commercially available *Lgr5* antibody for IF. This is the reason we resorted to use *Olfm4*, a robust marker for detecting the *Lgr5*⁺ intestinal stem cell population (**Extended data Figures 4b-c**).

Minor points:

This reviewer wonders why the authors do not clearly state the gene names encoding HP1 γ and lamin A. This information will help broader readers of Nature Communications.

These points have been stated:

Lanes 84-87: These observations, suggesting a role for HP1 γ in chronic inflammation, prompted us to generate a Villin-creERT2:*Cbx3*^{-/-} mouse model, allowing inducible inactivation of the *Cbx3* gene (encoding the HP1 γ protein) in the epithelial lineage of the gut.

Lanes 262-266: The *LmnA* gene includes 12 exons and, by alternative splicing, it will produce both lamin A and lamin C mRNA. Occasionally, a rare splicing event will also result in the production of progerin, a truncated version of lamin A acting as a dominant negative protein isoform responsible for the Hutchinson Gilford Progeria Syndrome (HGPS)

Table legends are missing in the PDF file.

Legends in Supplementary data Tables have been provided in the caption. Caption table description has now been introduced in the core of the manuscript.

There are some typos (l1139; Edu, l1289;potentiel).

These typos have been corrected

laminA and lamin A should be unified.

This has been modified as followed “lamin A”

Table 9 show exactly the same number of variants, such as Ugt1a7c, Ugt1a8, and Ugt1a9. This reviewers suspect this software cannot recognize the small differences of the sequence in the gene.

The Reviewer is right. We apologize for this mistake that has now been corrected.

Bibliography

1. Osorio, F. G. *et al.* Splicing-directed therapy in a new mouse model of human accelerated aging. *Sci Transl Med* **3**, 106ra107 (2011).
2. Tan, Q. *et al.* Extensive cryptic splicing upon loss of RBM17 and TDP43 in neurodegeneration models. *Human Molecular Genetics* **25**, 5083–5093 (2016).
3. Tréton, X. *et al.* Combined NADPH oxidase 1 and interleukin 10 deficiency induces chronic endoplasmic reticulum stress and causes ulcerative colitis-like disease in mice. *PLoS ONE* **9**, e101669 (2014).
4. Zong, X., Fu, J., Xu, B., Wang, Y. & Jin, M. Interplay between gut microbiota

- and antimicrobial peptides. *Anim Nutr* **6**, 389–396 (2020).
5. Zhang, X. *et al.* Splicing factor Srsf5 deletion disrupts alternative splicing and causes noncompaction of ventricular myocardium. *iScience* **24**, 103097 (2021).
 6. Fong, L. G. *et al.* Activating the synthesis of progerin, the mutant prelamin A in Hutchinson-Gilford progeria syndrome, with antisense oligonucleotides. *Human Molecular Genetics* **18**, 2462–2471 (2009).
 7. Vautrot, V. *et al.* Enhanced SRSF5 Protein Expression Reinforces Lamin A mRNA Production in HeLa Cells and Fibroblasts of Progeria Patients. *Hum Mutat* **37**, 280–291 (2016).
 8. Chai, G. *et al.* Mutations in Spliceosomal Genes PPIL1 and PRP17 Cause Neurodegenerative Pontocerebellar Hypoplasia with Microcephaly. *Neuron* **109**, 241–256.e9 (2021).
 9. Wen, J. *et al.* Zc3h13 Regulates Nuclear RNA m6A Methylation and Mouse Embryonic Stem Cell Self-Renewal. *Mol. Cell* **69**, 1028–1038.e6 (2018).
 10. Xiao, W. *et al.* Nuclear m(6)A Reader YTHDC1 Regulates mRNA Splicing. *Mol. Cell* **61**, 507–519 (2016).
 11. Haussmann, I. U. *et al.* m6A potentiates Sxl alternative pre-mRNA splicing for robust *Drosophila* sex determination. *Nature* **540**, 301–304 (2016).
 12. Zhao, Z. *et al.* N6-Methyladenosine RNA Methylation Regulator-Related Alternative Splicing (AS) Gene Signature Predicts Non-Small Cell Lung Cancer Prognosis. *Front Mol Biosci* **8**, 657087 (2021).
 13. Lopez-Mejia, I. C. *et al.* Antagonistic functions of LMNA isoforms in energy expenditure and lifespan. *EMBO Rep.* **15**, 529–539 (2014).
 14. Xiong, Z.-M., LaDana, C., Wu, D. & Cao, K. An inhibitory role of progerin in the gene induction network of adipocyte differentiation from iPS cells. *Aging (Albany NY)* **5**, 288–303 (2013).
 15. Cicero, Lo, A. *et al.* A High Throughput Phenotypic Screening reveals compounds that counteract premature osteogenic differentiation of HGPS iPS-derived mesenchymal stem cells. *Sci Rep* **6**, 34798–11 (2016).
 16. Buchwalter, A. & Hetzer, M. W. Nucleolar expansion and elevated protein translation in premature aging. *Nat Commun* **8**, 328–13 (2017).
 17. Hernandez-Segura, A. *et al.* Unmasking Transcriptional Heterogeneity in Senescent Cells. *Curr Biol* **27**, 2652–2660.e4 (2017).
 18. Kim, H., Choi, J. D., Kim, B.-G., Kang, H. C. & Lee, J.-S. Interactome Analysis Reveals that Heterochromatin Protein 1 γ (HP1 γ) Is Associated with the DNA Damage Response Pathway. *Cancer Res Treat* **48**, 322–333 (2016).
 19. Salton, M., Voss, T. C. & Misteli, T. Identification by high-throughput imaging of the histone methyltransferase EHMT2 as an epigenetic regulator of VEGFA alternative splicing. *Nucleic Acids Res* **42**, 13662–13673 (2014).
 20. Ameyar-Zazoua, M. *et al.* Argonaute proteins couple chromatin silencing to alternative splicing. *Nat. Struct. Mol. Biol.* **19**, 998–1004 (2012).
 21. Haberman, Y. *et al.* Ulcerative colitis mucosal transcriptomes reveal mitochondriopathy and personalized mechanisms underlying disease severity and treatment response. *Nat Commun* **10**, 38–13 (2019).
 22. Czarnewski, P. *et al.* Conserved transcriptomic profile between mouse and human colitis allows unsupervised patient stratification. *Nat Commun* **10**, 2892 (2019).
 23. Häsler, R. *et al.* Alterations of pre-mRNA splicing in human inflammatory bowel disease. *Eur J Cell Biol* **90**, 603–611 (2011).
 24. Häsler, R. *et al.* Uncoupling of mucosal gene regulation, mRNA splicing and

adherent microbiota signatures in inflammatory bowel disease. *Gut* **66**, 2087–2097 (2017).

REVIEWER COMMENTS

Reviewer #1 (Remarks to the Author):

The authors addressed my comments satisfactorily.

Reviewer #3 (Remarks to the Author):

The revised manuscript is significantly improved, but still fail to clarify the critical point. HP1 α , β , and γ are known to contribute to transcription regulation as well as heterochromatin formation and maintenance. Therefore, it is possible that the phenotypes of Cbx3 KO mice result from not just splicing noise, but/and genome wide alteration of heterochromatin state (PMID: 27090491, 34194047). Since HP1 α and β are compensatory upregulated in Cbx3 KO epithelia as previously shown, heterochromatin regions are supposed to be affected or re-distributed in Cbx3 KO mice. The authors should evaluate the alteration of global heterochromatin state. To this aim, it is not sufficient to assess the net deposition of H3K9me₃; the genome-wide correlation between HP1 γ and H3K9me₃ should be assessed in WT epithelia and global heterochromatin state, H3K9me₃ deposition, in WT and Cbx3 KO epithelia should be compared.

In addition, this reviewer agrees with importance of splicing noise. However, it is still possible that interference with LMNA contributes to the biology independent of Cbx3 deficiency. The authors should analyze more highly affected genes than Lmna.

There remain some typos (e.g. "Ileon").

Response to Reviewer's comments

Reviewer #1 (Remarks to the Author):

The authors addressed my comments satisfactorily.

Reviewer #3 (Remarks to the Author):

The revised manuscript is significantly improved, but still fail to clarify the critical point.

HP1 α , β , and γ are known to contribute to transcription regulation as well as heterochromatin formation and maintenance. Therefore, it is possible that the phenotypes of Cbx3 KO mice result from not just splicing noise, but/and genome wide alteration of heterochromatin state (PMID: 27090491, 34194047).

Since HP1 α and β are compensatory upregulated in Cbx3 KO epithelia as previously shown, heterochromatin regions are supposed to be affected or re-distributed in Cbx3 KO mice. The authors should evaluate the alteration of global heterochromatin state. To this aim, it is not sufficient to assess the net deposition of H3K9me₃; the genome-wide correlation between HP1 γ and H3K9me₃ should be assessed in WT epithelia and global heterochromatin state, H3K9me₃ deposition, in WT and Cbx3 KO epithelia should be compared.

We would first like to thank the referee for acknowledging that the revised manuscript was improved.

We agree with the Reviewer that change in heterochromatin state can contribute to gene de-repression in Cbx3 KO mice. However, this de-repression is not global as we now provide additional evidence documenting that neither pericentromeric nor interspersed heterochromatin become destabilized in the absence of HP1 γ .

First, immunocytochemistry experiments at higher resolution document that Cbx3 inactivation does not result in disorganization of the overall structure of heterochromatin. Notably, we show that the chromocenters are preserved and that association of the heterochromatic marks H3K9me₃ and H4K20me₃ with these regions is unaffected (**Extended data Figures 1 and 2**). This is in contrast to an earlier study reporting that these histone marks show reduced association with chromocenters upon HP1 α /HP1 β /HP1 γ triple knock-out in hepatocyte (PMID: 32020053).

Next, we have now also examined the effect of Cbx3 inactivation on endogenous retroviruses (ERVs) using our transcriptomic data (**Extended data Figure 7a-c**). Earlier studies have shown that these DNA repeats are normally kept in check by H3K9 trimethylation in the gut epithelium (PMID: 32503845, PMID: 32296174). We find no significant upregulation of any of the more abundant families of ERVs in any of the 3 tissues (crypt, villi and colon epithelia). By contrast, families of ERVs were reactivated in triple knock-out of HP1 α , HP1 β , and HP1 γ in mouse liver (**Extended data Figure 7d**, transcriptome data from). Together, these observations document that inactivation of more than just Cbx3 is more likely required to destabilize interspersed heterochromatin.

Finally, we would like to draw the attention of the Reviewer on the time scale required for the suggested *in vivo* ChIP-seq experiments with HP1 γ and H3K9me₃, as breeding the mice of the required genotype would take months (for example, PMID: 26881867 used pooled 7 to 32 mice crypt/villi epithelia for the acquisition of meaningful Chip-Seq peaks), while, as argued above, we may not be certain that the experiment will be informative beyond existing data.

Thus, the new version includes:

- representative IF experiments showing the distribution of the heterochromatic marks H3K9me3 and H4K20me3.
- an analysis of ERV expression per family, showing no significant up-regulation of major clades of endogenous retroviruses upon inactivation of Cbx3 (using the triple knock-out of HP1 α , HP1 β , and HP1 γ in liver as a positive control).

In addition, this reviewer agrees with importance of splicing noise. However, it is still possible that interference with LMNA contributes to the biology independent of Cbx3 deficiency.

In the revised version, the introduction and the discussion have been corrected to make even clearer in the that we do not claim that the entirety of the KO mouse phenotype is explained by splicing noise, and that gene de-repression is also an important driver of this phenotype.

The authors should analyze more highly affected genes than Lmna.

We have now carried out RT-qPCR experiments validating several of the splicing events predicted as altered by analysis of the RNA-seq data (**Extended data Figure 9 and 10**). These experiments also document that PCR products are frequently more complex upon Cbx3 inactivation as a consequence of the increased splicing noise.

REVIEWERS' COMMENTS

Reviewer #3 (Remarks to the Author):

First of all, we thank the authors for their efforts provided in this revised edition. The new extended Fig. 7 has completely resolved the ambiguity of the location-specific heterochromatin status within the crypts.

However, all data on ERV transcript levels and fluorescence intensities of H3K9me3 and H3K20me show no difference in the net heterochromatin state between CBX3 KOs and controls, and the possibility of redistribution of heterochromatin sites without global loss or gain of heterochromatin state cannot be excluded (PMID: 25938714). Without specific or global H3K9me3 and/or H3K20me3 status at the progerin locus or beyond, the possibility that HP1 γ inactivation substantially affects heterochromatin integrity should not be excluded.

Extended Fig. 2; fluorescence intensities of H3K9me3 and H3K20me need to be quantified.

Response to Reviewer's comments

Reviewer #3 (Remarks to the Author):

First of all, we thank the authors for their efforts provided in this revised edition. The new extended Fig. 7 has completely resolved the ambiguity of the location-specific heterochromatin status within the crypts.

However, all data on ERV transcript levels and fluorescence intensities of H3K9me3 and H3K20me show no difference in the net heterochromatin state between CBX3 KO and controls, and the possibility of redistribution of heterochromatin sites without global loss or gain of heterochromatin state cannot be excluded (PMID: 25938714). Without specific or global H3K9me3 and/or H3K20me3 status at the progerin locus or beyond, the possibility that HP1 γ inactivation substantially affects heterochromatin integrity should not be excluded.

This statement has been now introduced in the discussion

Extended Fig. 2; fluorescence intensities of H3K9me3 and H3K20me need to be quantified.

The new version of the manuscript includes the quantification of the fluorescence intensities as Supplementary Figures 1c, 1e and Supplementary Figures 2b and 2d